



# Aerosol uncertainties in tropical precipitation changes for the mid-Pliocene Warm Period

Anni Zhao[1,2], Ran Feng[3], Chris M. Brierley[2], Jian Zhang[1], and Yongyun Hu[1]

[1]Department of Atmospheric and Oceanic Sciences, School of Physics, Peking University
[2]Department of Geography, University College London
[3]Center for Integrative Geosciences, University of Connecticut

**Correspondence:** Anni Zhao (annizhao@pku.edu.cn) and Yongyun Hu (yyhu@pku.edu.cn)

**Abstract.** The mid-Pliocene Warm Period (mPWP, 3.3 - 3.0 Ma) was characterised by an atmospheric $CO_2$ concentration exceeding 400 ppmv with minor changes in continental and orbital configurations. Simulations of this past climate state have improved with newer models, but still show some substantial differences from proxy reconstructions. There is little information about atmospheric aerosol concentrations during the Pliocene, but previous work suggests that is could have been quite
different from the modern. Here we apply idealised aerosol scenario experiments to examine the importance of aerosol forcing on mPWP tropical precipitation and the possibility of aerosol uncertainty explaining the mismatch between reconstructions and simulations. The absence of industrial pollutants leads to further warming especially in the Northern Hemisphere. The Intertropical Convergence Zone (ITCZ) becomes narrower and stronger and shifts northward after removal of anthropogenic aerosols. Though not affecting the location of monsoon domain boundary, removal of anthropogenic aerosol alters the amount
of rainfall within the domain, increasing summer rain rate over eastern and southern Asia and western Africa. This work demonstrates that uncertainty in aerosol forcing could be the dominant driver in tropical precipitation changes during the mid-Pliocene: causing larger impacts than the changes in topography and greenhouse gases.

## 1 Introduction

The mid-Pliocene or mid-Piacenzian Warm Period (mPWP, around 3.3-3.0 Ma, Dowsett et al., 1999; Haywood et al., 2016a) was the most recent time in Earth's history when atmospheric $CO_2$ concentration exceeded 400 ppmv (Bartoli et al., 2011; de la Vega et al., 2020) with nearly the same topography and orbital configuration as modern. This period is recognised as the last period with quasi-equilibrium warm climate before Pleistocene when climate had nearly fully responded to the high atmospheric $CO_2$ concentration (e.g. Haywood et al., 2010). Understanding the environmental process in the mPWP provides a
chance to understand how the climate system would respond to a perturbation of radiative forcing and to evaluate the impacts of climate change caused by great $CO_2$ emissions (e.g. Haywood et al., 2016b; Pagani et al., 2010). The first study reconstructing regional Pliocene climate was conducted by Zubakov and Borzenkova (1988) based on more than 20 sequences from terres-



trial and marine cores. They pointed out that the Pliocene Optimum could be a future analogue, with high $CO_2$ concentration and climate showing 100-300 kyr cycles with a 4-5 °C amplitude. Proxy data suggested a warming in global mean surface temperature (GMST) of 2.5 - 4.0°C during the mPWP than 1850-1900 CE, based on the estimation of global sea surface temperature (GSST) anomaly and its relationship in changing rate with the GMST (Gulev et al., 2021). The Pliocene climate was characterized by reduced temperature gradients (Haywood et al., 2013, 2020; Dowsett et al., 2013; Foley and Dowsett, 2019) from equator to pole e.g.(e.g. Dowsett and Poore, 1991) and from western to eastern tropical Pacific Ocean which has been recognised as a tropical Pacific warm pool pattern (Wara, 2005; Dowsett, 2007; Zhang et al., 2014; O'Brien et al., 2014).

Climate models have been used to simulate mPWP climate and to investigate the drivers and mechanisms behind the mPWP climate changes shown in proxy data (e.g. Fedorov et al., 2013). The Pliocene Model Intercomparison Project (PlioMIP; Haywood et al., 2010, 2016b) is a coordinated international climate modelling project initiative aimed of understanding the climate and environments of mPWP, exploring model uncertainties and evaluating the potential relevance to future climate change. It estimates a warming of 3.2°C (2.1°C to 4.8°C) compared to the pre-industrial control runs in the PlioMIP2 (Haywood et al., 2020) and 2.7°C (1.8°C to 3.6°C) in the PlioMIP1 (Haywood et al., 2013). PlioMIP simulations suggest enhanced monsoons in West Africa, India and East Asia (Zhang et al., 2013; Berntell et al., 2021; Han et al., 2021; Feng et al., 2022), and precipitation in the South Pacific Convergence Zone (SPCZ) and the ITCZ during the mPWP (Haywood et al., 2020; Han et al., 2021). Yet ongoing climate change since the $20^{th}$ century has not shown those mPWP precipitation changes during which monsoon precipitation is affected by the counteraction between Greenhouse Gases (GHGs) induced increase and emission induced decrease (Douville et al., 2021). Many climate features, especially the hydroclimate features, of the mPWP differ substantially from the projected features of near future climate (Feng et al., 2022; Bhattacharya et al., 2022; Burton et al., 2023). Although the mPWP climate was thought to reflect earth system responses that operates at a longer time scale (Lunt et al., 2010; Feng et al., 2022), Pliocene climate was also likely driven by different atmospheric aerosol forcing conditions (Unger and Yue, 2014; Sagoo and Storelvmo, 2017). Whether those different forcing conditions play an role in driving the Pliocene hydroclimate remains unknown.

Aerosols are an important driver in climate system. Knowing their atmospheric concentrations and mechanisms behind climate response are key tasks for simulating past and future. Aerosols directly affect absorbed incoming solar radiation and indirectly affect the radiative flux through their role in determining cloud properties. Aerosol-cloud interactions include the changes in cloud albedo through altering droplet size and concentration (Twomey, 1977) and changes in cloud lifetime through altering the precipitation efficiency (Albrecht, 1989). Future changes in the global aerosol burden are dependent on anthropogenic emissions, yet natural sources of aerosol emissions are also expected to change (Tegen and Schepanski, 2018). mPWP simulations now use modern-day or pre-industrial aerosol concentration that may differ from the conditions during the mPWP. It implies that aerosol effects may be one of the possible explanation for the mismatch between reconstructions and simulations. However, little research has tried to understand aerosol effects in the Pliocene (e.g. Unger and Yue, 2014). More attention in palaeoclimate studies relevant to aerosols has been paid to the Last Glacial Maximum (LGM) period when the global dust cycle was enhanced (Lambert and Albani, 2021). An idealised study suggests the importance of aerosol and chemistry-climate feedbacks in modelling Pliocene climate, as the aerosol cooling compensates 15-100 % of the warming induced by high $CO_2$





while chemistry-climate feedback warms the climate with the magnitude of 30-250% of the $CO_2$ induced warming (Unger and Yue, 2014). Sagoo and Storelvmo (2017) found that dust indirect effects could explain some of the mismatch between model and data for LGM and mPWP. They used a new empirical parameterisation for ice nucleation on dust particles to investigate radiative forcing induced by different dust loading from low to high. Results showed that increased dust reduces the size of ice crystals in clouds while increasing their amount. This increases the shortwave cloud radiative forcing, thus, cools surface temperature and amplifies polar warmth (Sagoo and Storelvmo, 2017).

In this study, we analyse two Pliocene simulations with different aerosol scenarios to further investigate the potential effect of aerosol on mPWP climate, one with pre-industrial aerosol concentrations and one with present-day aerosol concentrations published in Lamarque et al. (2010). The simulations were performed initially to analyse the effects of aerosol-cloud interactions on mPWP seasonally sea ice-free Arctic (Feng et al., 2019). We compare the changes caused by removing anthropogenic concentrations in Pliocene, in order to raise the importance of considering aerosol effects in modelling Pliocene climate. This investigation also highlights the potential analogue nature of Pliocene climate, not to the present-day polluted climate, but to future climate scenarios which feature removal of anthropogenic pollutants. We focus on changes in tropical precipitation, because precipitation varies much more than other near-uniform variables like temperature and irradiation in the tropics.

## 2 Methods

### 2.1 Model description

Simulations were performed with the Community Earth System Model version 1.2 (CESM1.2; Hurrell et al., 2013), consisting of Community Atmospheric Model version 5.3 (CAM5.3; Martinez, 2012), Parallel Ocean Program version 2 (POP2; Danabasoglu et al., 2012), Community Land Model version 4 (CLM4; Oleson, 2010) and Community Ice Code version 4 (CICE4; Holland et al., 2012). Differences and similarities between CESM1 and CCSM4 are summarised in Meehl et al. (2013). The key differences between CCSM4 and CESM1 come from a new aerosol scheme, inclusion of aerosol-cloud-interactions, and a more realistic boundary layer and radiation in CAM5 (Meehl et al., 2013; Martinez, 2012). The 3-mode version of modal aerosol module (MAM3; Liu et al., 2012) is used as the aerosol micro-physical scheme for long-term climate simulations, which uses Aiken, accumulation and coarse modes to solve number and size concentration of internal condensation and coagulation of different species among modes. Assumptions and limitations of MAM3 are described in Liu et al. (2012). Though CAM5 simulates too strong response in cloud radiative to aerosol changes (Martinez, 2012), its ability to simulate cloud and cloud radiative forcing has been improved since the previous generations (Kay et al., 2012).

### 2.2 Experimental design

The simulations were branched from an existing CCSM4-PlioMIP1 simulation that only considers the direct effect (Rosenbloom et al., 2013). The resolution was set up to 0.9° x 1.25° for the atmospheric and land components and ~1° for the oceanic





and sea ice components. Two pollutant scenarios were applied. One simulation applied pre-industrial emissions (hereafter re-
ferred as to Plio_Pristine, which can be treated as equivalent to a PlioMIP1 simulation). The other prescribed pre-industrial
emissions plus industrial pollutants of anthropogenic $SO_2$, sulfate and organic compounds estimated for the 2000s from an
gridded (0.5° x 0.5°) emission dataset (hereafter referred as to Plio_Polluted) published in Lamarque et al. (2010). Compared
to Plio_Polluted, Plio_Pristine simulates a positive adjusted effective radiative forcing of 1.29 W m$^{-2}$, in which radiative
forcing due to aerosol cloud interactions contributes to 1.11 W m$^{-2}$ and direct radiative forcing contributes to 0.18 W m$^{-2}$
(Feng et al., 2019). The emission dataset (Lamarque et al., 2010) includes reactive gases and aerosols covering the historical
period from 1850 to 2000, with the aim of providing consistent gridded emissions for CMIP5 models to run chemistry model
simulations that would contributed to the assessment in IPCC AR5. Boundary conditions were set up based on the PlioMIP1
(Haywood et al., 2011). Each simulation was branched from model year 500 of the CCSM4-PlioMIP1 simulation, and run for
another 300 model years. Outputs are taken as the averaged means of the last 50 model years.

**2.3    Analysis techniques**

**2.3.1    Monsoon analysis**

It is not appropriate to use fixed present-day monsoon domain boundary to analyse paleo monsoon properties as the monsoon
domain varies through time. Here we adopt the definition of monsoon domain in Wang and Ding (2008) and Wang et al.
(2014). Local summer is defined as May-September (MJJAS) in the Northern Hemisphere or November-March (NDJFM) in
the Southern Hemisphere, and local winter is defined as NDJFM in the Northern Hemisphere and MJJAS in the Southern
Hemisphere. Averaged summer rain rate takes the averaged daily rain rate during local summer. Monsoon intensity is the
difference in averaged daily precipitation rate between local summer and local winter. Global monsoon domain is defined as at
least of 55% of the annual total rainfall comes from summer, and the monsoon intensity is no less than 2 mm per day.

**2.3.2    Data model comparison**

Simulated mPWP annual mean temperature anomalies (Plio_Pristine - PI) are compared to the reconstructed mPWP sea surface
temperature (SST) anomalies, which were presented as the reconstructed mPWP SSTs dataset in PRISM4 (Foley and Dowsett,
2019) with an interval of 30,000 years minus the observed 1870-1988 SSTs from the NOAA Extended Reconstructed Sea
Surface Temperature (ERSST) version 5 dataset (Huang et al., 2017). Salzmann et al. (2008) compared 28 present-day mean
annual precipitation anomalies (mid-Pliocene - present-day) from literature for selected regions. Only 10 data points were
115 chosen for data-model comparison in this study as their precise coordinates and anomalies could be found in literature. The
10 points have been regenerated into 6 after the combination of points if their sites were located within a single model grid.
Three sites in western coast of USA and 3 in Yunnan, China have been regenerated into new sites respectively by taking their
averages. Therefore each three have been reorganised into a single point by taking their average. (Feng et al., 2022) published
a compilation of mPWP proxy records reflecting the mPWP terrestrial hydroclimate change shown in the difference between
120 precipitation and evaporation, which consists of 62 data points after the exclusion by quality control. Here we follow the





treatment of (Feng et al., 2022) to combine the co-located sites if they were less than 150 km apart and featured same-signed anomalies. The combination regenerates the site sets 35, 72; 22, 77; 79,61; 58,61,79,101 and 65, 66 into new sites located at the center of sets.

## 3 Results

### 3.1 Simulated mPWP climate change relative to the pre-industrial

Fig. 1a shows that the tropical annual mean surface temperature of Plio_Pristine is overall warmer than the pre-industrial (hereafter referred as to PI). Plio_Pristine produces warmer high-latitude Pacific and Southern Ocean than the tropics as compared to PI. Eastern Pacific Ocean along the tropics and the upwelling region along the coast line of North America is about 0.5°C warmer than western Pacific. Plio_Pristine captures both reduced meridional and zonal temperature gradients during the mPWP that have been shown in the PlioMIPs (Haywood et al., 2013, 2020), but the amplifying magnitude in Plio_Pristine is weaker than the PlioMIPs (Fig. 2a). The underestimation in Northern Hemisphere warming may be explained by that Plio_Pristine was branched from an earlier CCSM4-PlioMIP1 simulation that had underestimated the warming in the Northern Hemisphere (Rosenbloom et al., 2013). Though Plio_Pristine underestimates positive temperature anomalies, its simulated change still within the range suggested by the PlioMIPs. 13 out of the 37 sites show a mismatch between simulated surface temperature anomaly and reconstructed SST anomaly smaller than 1.0°C. Large mismatch occurs in high-latitudial North Atlantic Ocean and the upwelling region near the western coastline of South Africa. Reconstructed SST anomalies show an averaged warming of 4.4°C north to 40°N of North Atlantic Ocean and 7.0°C over the upwelling region, which are 3.5°C and 5.6°C higher than the simulated warming, respectively. The reconstructed zonal temperature gradient along the tropical Pacific Ocean is around 0.5°C greater than simulations (Plio_Pristine - PI). Data model mismatch suggests that Plio_Pristine underestimates the magnitude of reduction in both SST gradients during the mPWP as compared to reconstructed SST anomalies (Foley and Dowsett, 2019).

Previous studies of mPWP monsoon change based on data were mainly focused on Asian monsoons due to the rare availability of geological evidence, which suggested a wetter East Asian monsoon during the mPWP (e.g. Xiong et al., 2010; Nie et al., 2014). The PlioMIPs show enhanced monsoon over West and North Africa, India and East Asia that are consistent with proxy data (Li et al., 2018; Feng et al., 2022). Fig. 1b shows that comparing to the PI, Plio_Pristine produces greater annual mean precipitation (as the sum of stable large-scale and convective precipitation rate) over tropical oceans and land monsoon regions (western and northern Africa, East Asia, Australia and parts of Southeast Asia). Precipitation decreases over subtropical oceans, eastern parts of South America and monsoon areas over North America. Plio_Pristine produces the right sign of precipitation as proxy data suggest, but in general underestimates the enhancing magnitudes especially over Australian sites where proxy data suggest large positive precipitation anomaly (Fig. 1b and 2b). Though Plio_Pristine produces less precipitation over tropical Northern Hemisphere as compared to the multi-model means from PlioMIP1 (Haywood et al., 2013) and PlioMIP2 (Haywood et al., 2020), it overall agrees with the findings from the PlioMIPs.



### 3.2 Effects of removing anthropogenic aerosols on temperature

Reducing atmospheric aerosols is expected to warm the climate. Removal of present-day anthropogenic aerosols from the atmosphere (Plio_Pristine - Plio_Polluted) causes a global warming of 0.84 °C (Feng et al., 2019), which is close to the estimated modern cooling contributed by aerosols at 0 - 0.8 °C in the IPCC AR6 (IPCC, 2021). Simulated warming is in line with the the theory of aerosol effects, as aerosols contribute a negative forcing on climate which in turn removing aerosols from the atmosphere should lead to a positive forcing.

Spatial pattern of temperature change is consistent with the change in energy flux. Overall, asymmetrical change in radiation flux matches with the larger aerosol concentration change in the Northern Hemisphere (Fig. 3a and 4a). Major change in energy flux after the removal of anthropogenic emissions occurs in the Northern hemisphere especially over lands that leads to further warming. The warming pattern due to asymmetrical emissions concentrating in the Northern Hemisphere agrees with the findings from an earlier study (Samset et al., 2018), in which several models have been applied to test removing all

anthropogenic emission under present-day conditions. Annual mean temperature rises about 0.4 - 1.0 °C in tropical Pacific Ocean and causes more warming by increasing 1.0 - 1.4 °C in subtropical Pacific and the upwelling region in eastern Pacific after removing pollutants (Fig. 5a), which therefore reduces both meridional and zonal temperature gradients in Pacific Ocean. In future climate projections, removing emitted pollutants is shown to enhance global mean warming, especially in the Northern Hemisphere where the emissions mostly come from. However, removing anthropogenic aerosols causes more warming than

the mPWP boundary conditions over Northeastern Pacific and high-latitude North Atlantic Ocean (Fig. 5a,c,e). This suggests that changes in boundary conditions are relative more important than aerosol effects in mPWP warming.

### 3.3 Effects of removing anthropogenic aerosols on tropical precipitation

Precipitation responds to removal of aerosols in a more complex manner, but in general it responds intensively in deep convective regions (Fig. 5b,d,f). The precipitation change pattern matches more with mid- and high- level clouds (Fig. 6 and

175 A2. Precipitation enhances in the tropics while reduces in the adjacent subtropics in response to removing emissions, which shows that the local convection becomes narrower and stronger with a northward shift. The narrower ITCZ is consistent with a warmer climate simulated by Plio_Pristine, and agrees with an earlier study which suggests that future ITCZ would become narrower and weaker under global warming in the CMIP5 simulations (Byrne and Schneider, 2016). Ridley et al. (2015) and Voigt et al. (2017) found that introducing anthropogenic aerosols causes southward shift of ITCZ. In our case, the northward

shift of ITCZ due to removal of anthropogenic aerosols is consistent with this finding. The pattern of the magnitudes of tropical precipitation anomaly matches the tropical temperature warming over oceans, which agrees with Berg et al. (2013) as higher temperatures cause stronger convective precipitation. Tropical precipitation increases greater over the tropical Pacific and Indian Ocean where the warming is stronger than the tropical mean warming (Fig. 5), consistent with the positive correlation between tropical precipitation change and spatial deviations of SST warming from the tropical mean (Xie et al., 2010). In the

northern tropics, anthropogenic aerosols caused a rainfall reduction through the twentieth century (Ridley et al., 2015). Correspondingly, precipitation should increase after removal of emissions. The positive precipitation change over Asia between



Plio_Pristine and Plio_Polluted agrees with the argument.

Strong precipitation response to removal of anthropogenic aerosols implies that the choice of prescribed aerosol scenario could affect simulated mPWP monsoon response. Figure 7 shows the change in global monsoon domain and summer rain rate between Plio_Pristine and PI (panel a) and between Plio_Pristine and Plio_Polluted (panel b). Little difference among the boundary of global monsoon domain for the PI, Plio_Pristine and Plio_Polluted implies that removal of anthropogenic aerosols has little effect on the location of global monsoon domain. Though not affecting locations, aerosol forcing alters the spatial distribution of land monsoon precipitation. Land regional monsoons are enhanced, agreeing with the observations that reduce aerosol-induced cooling and increase the precipitation over western Africa and eastern and southern Asia during local monsoon seasons as anthropogenic emission partly caused the global land monsoon precipitation reduction during 1950s to 1980s and weakened western African and eastern and southern Asian monsoon over the 20th century by cooling effect (Douville et al., 2021). Comparison between Plio_Pristine and Plio_Polluted shows the expected change in temperature and monsoon (Fig. 5,7), which is consistent with projected future monsoon enhancement, as global and Asian summer monsoon precipitation is likely to increase by 2050s due to the expected reductions in anthropogenic aerosol emissions (Wilcox et al., 2020). Though this study used idealised aerosol scenario that could not occur during mPWP, it reveals the importance of prescribed aerosol scenario on simulating mPWP monsoon response.

## 4 Discussion

### 4.1 Key forcing driving tropical precipitation change

Aerosols affect the climate directly by altering radiation and indirectly by interacting with clouds via changing cloud albedo and cloud lifetime. In tropical regions, cloud radiative effects primarily drive the energy flux change after removing pollutants (Fig. 3). Aerosols play less role in absorbing shortwave radiation in the Southern Hemisphere. Clear-sky net shortwave flux gradually increases towards north, but its magnitude is much smaller than the full-sky condition, while clouds contribute to a net forcing greater than 2 $\mathrm{Wm}^{-2}$ with more shortwave absorption and less longwave emission. In the Northern Hemisphere, aerosol reflection reduces consistently with the decrease in aerosol optical depth (Fig. 4a), but is still less important than the change in shortwave cloud forcing. Feng et al. (2019) focused on change in cloud droplet properties, as shortwave cloud forcing is dependent on cloud optical depth, whose change is positively correlated with the change in cloud droplet concentration and cloud liquid water content. Figure 6 shows decrease in droplet concentration (especially over lands and high latitudes, Fig. A3a) and decrease in cloud liquid path (except tropical Pacific between 5°S and 5°N and western Africa, Fig. A3b) after the removal of pollutants. The uniform decrease in droplet concentration and overall decrease in cloud liquid path suggests lower cloud albedo and shorter cloud lifetime due to removing anthropogenic aerosols.



### 4.2 Relative importance comparison between mPWP boundary conditions and aerosol forcing

The overall uniform warming over tropical and subtropical regions (Plio_Pristine - PI and Plio_Pristine - Plio_Polluted) result in general increase in sea level pressure and have relatively small effects on surface wind, while removing anthropogenic aerosols rises the sea level pressure and have much stronger effects on surface wind that shows seasonal variances (Fig. 9). Studies have highlighted that anthropogenic aerosols cause the southward shift in the ITCZ, weaken the Hadley circulation and reduce the precipitation in deep convective areas in response to the Northern Hemisphere cooling induced by aerosols (Hwang et al., 2013; Wang et al., 2019). Removal of human-induced aerosol emissions causes more precipitation change than the mPWP boundary condition over subtropical Pacific Ocean and South Atlantic Ocean, Indian Ocean and Southeast Asia. Figure 10 shows the relative importance of removing anthropogenic aerosols and mPWP boundary conditions on mPWP zonal mean precipitation change. The ratio (panel a) indicates that the relative importance of effects of aerosol forcing and mPWP boundary conditions (including high $CO_2$) on precipitation is complicated, but overall aerosol effect is more important over the tropics (panel b), which could imply the importance of aerosol scenario in simulating Pliocene climate. It highlights the importance of aerosol in mPWP data model comparison, as proxy recorded tropical temperature and hydroclimate conditions can be sensitive to aerosol emission conditions during the Pliocene.

### 4.3 Implications for simulating Pliocene climate

Strong tropical precipitation change in response to removal of anthropogenic aerosols indicates that simulated mPWP precipitation is sensitive to prescribed aerosol scenarios. Previous simulations of the mPWP atmospheric chemistry confirmed that there is a significant increase in the concentrations of aerosol precursors from terrestrial ecosystem emissions (Unger and Yue, 2014) and that simulated mPWP climate is sensitive to indirect effects of dust (Sagoo and Storelvmo, 2017). However, Unger and Yue (2014) only looked at direct radiative effects on mPWP climate induced by feedbacks from physical climate change on reactive composition, yet they did not consider aerosol-cloud interactions. To what extent the terrestrial biogenic aerosols may change mPWP hydrological cycle remains unknown. In Unger and Yue (2014), continental greening and warmer climate during the mPWP leads to greater emissions of aeorosols and precursors from more active wildfire and biome productions, yet, this increase is likely much smaller than the anthropogenic pollutant emissions. For example, due to the enhanced wildfire activity, $SO_2$ increased by 11 Tg in Unger and Yue (2014) mPWP simulations, yet $SO_2$ from anthropogic sources is estimated to be 120 Tg from 1980 to 2000s (Smith et al., 2011). As such, despite that enhanced biogenic emissions may compensate some of the responses seen in the Plio_Pristine, we argue that the absence of anthropogenic pollutant from the mPWP troposphere remains one of the important factors driving the differences between the mPWP and near future climate, especially the Arctic climate (Feng et al., 2019) and hydroclimate. In addition, the mPWP also likely featured different amount of dust aerosols. Sagoo and Storelvmo (2017) looked in to the effect of dust and demonstrated that increased dust emissions resulted in an enhanced dynamical response that lead to great changes in hydrological circulations. Their Pliocene simulation with idealised dust scenario shows greater change in precipitation than only consider $CO_2$ forcing, yet only used idealized scenario (Sagoo and Storelvmo, 2017). Usage of more realistic prescribed aerosol scenario should be considered in future work.





Finally, despite that we analyzed change in mPWP hydrological cycle, we didn't fully account for changes in the moisture budget that explains our simulated hydroclimate changes in the Plio_Prisine. The anomalous moisture budget can be decomposed in to thermodynamic, dynamic and residual components etc (Trenberth and Guillemot, 1995; Held and Soden, 2006). D'Agostino et al. (2019, 2020) state that monsoon changes under RCP8.5 are likely driven by thermodynamics and net energy input in response to global warming. As the high $CO_2$ concentration is also a major forcing in mPWP simulations, it implies that

thermodynamic change plays an important role in mPWP precipitation change as well. Removal of anthropogenic emissions in this work would logically increase the effects of thermodynamic and net energy input due to its induced further warming and increased net radiative absorption. Further study could focus on decomposing moisture budget to determine which component plays an important role in generating mPWP precipitation change.

## 5 Conclusions

A set of existing simulations with two idealised aerosol scenarios (Feng et al., 2019) are analysed in this study to investigate the importance of aerosol forcing on mPWP tropical climate. Our results highlight the importance of prescribed aerosol scenarios in simulating mPWP precipitation change. In contrast to GHGs warm the climate, aerosols cool the climate. As expected, removal of pollutants causes global warming and further warms the Northern Hemisphere due to asymmetrical emissions. Aerosols affect precipitation in a more complicated manner. Precipitation enhances in the tropics especially over deep convective regions

while reduces in adjacent subtropics showing a narrower and stronger with a northward shift in the ITCZ. Though not affecting the location of monsoon domain boundary, removal of anthropogenic aerosol changes the precipitation within the domain and increases summer rain rate over eastern and southern Asia and western Africa. Removal of anthropogenic emissions leads to lower cloud albedo and shorter cloud lifetime in simulated mPWP climate. Aerosols have more impacts on annual and seasonal precipitation over the tropics and subtropics than the mPWP boundary conditions. In reality, the concentration and composition

of aerosol during the mPWP were different to those in the pre-industrial and present-day, due to mPWP had warmer and wetter climate with greater vegetation cover and natural emissions without human effect, respectively. Climate response to aerosol forcing during the mPWP was expected to be different to those periods. It is important to consider the differences in aerosol conditions across the periods.




*Code and data availability.* The simulation data are hosted at Cheyenne supercomputer with the path: /CCSM/csm/CESM-CAM4-PLIOCENE. Instructions and permissions to relevant experiments will be provided upon request to ran.feng@uconn.edu. The processed data for figures are available at https://github.com/annizhao1994/mPWPAerosol and https://zenodo.org/doi/10.5281/zenodo.10130118. Source code of CESM1.2 can be downloaded from https://www2.cesm.ucar.edu/models/cesm1.2/


*Author contributions.* A.Z and C.M.B. performed the bulk of the writing and analysis. R.F contributed data for the manuscript and payed large effort on revision. Y.H and J.Z. revised the manuscript.

*Competing interests.* At least one of the co-author are members of the editorial board of Climate of the Past. The authors have no other
competing interests to declare.

*Acknowledgements.* We acknowledge the high-performance computing support from Cheyenne (https://doi.org/10.5065/D6RX99HX) and Yellowstone (ark:/85065/d7wd3xhc) provided by NCAR's Computational and Information Systems Laboratory, sponsored by the National Science Foundation. This material is based upon work supported by the National Center for Atmospheric Research, which is a major facility
sponsored by the National Science Foundation under Cooperative Agreement 1852977. R.F. received funded by NSF grant 2103055 and 2238875. Y.H were founded in part by the National Natural Science Foundation of China, under grant 41888101. J.Z. was founded by National Natural Science Foundation of China grant 42372120.



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

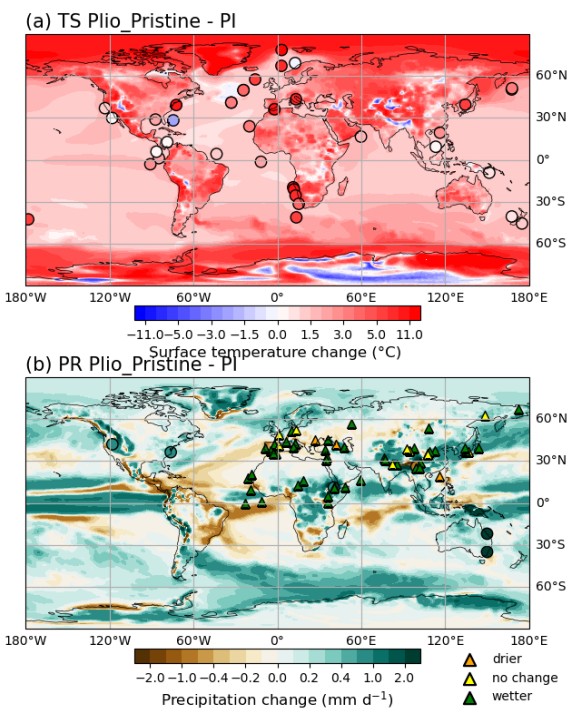

**Figure 1. Changes in (a) mean annual surface temperature (TS) in °C and (b) precipitation (PR) in mm d$^{-1}$ (Plio_Pristine - PI).**
Shaded circles are reconstructed anomalies of (a) SSTs (Foley and Dowsett, 2019) and (b) precipitation (Salzmann et al., 2008). Triangles mark the sign of mPWP hydroclimate changes (mean annual precipitation minus evaporation) published in (Feng et al., 2022). See section 2.3.2 for details.

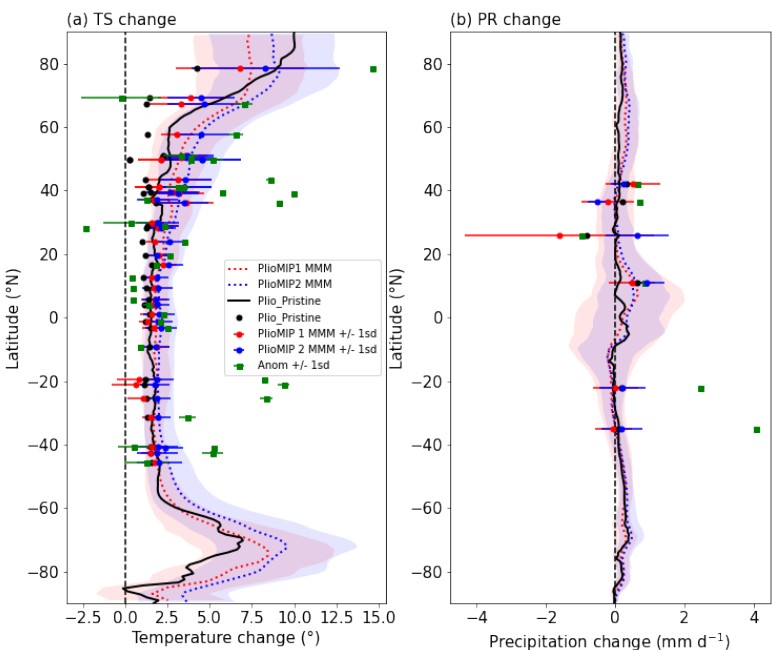

**Figure 2. Zonal averaged changes in mean annual (a) surface temperature in °C and (b) precipitation in mm d⁻¹.** In all panels, black solid lines show the simulated anomalies (Plio_Pristine - PI). Dotted lines represent the multi-model means of the PlioMIP1 (red, Haywood et al., 2013) and PlioMIP2 (blue, Haywood et al., 2020) respectively, with shaded areas showing the model spread, i.e. the standard deviation of the ensemble. Green squares show the site-level reconstructions in Fig. 1, and dots show the corresponding grid-level anomalies with error bars showing the range of the PlioMIPs.





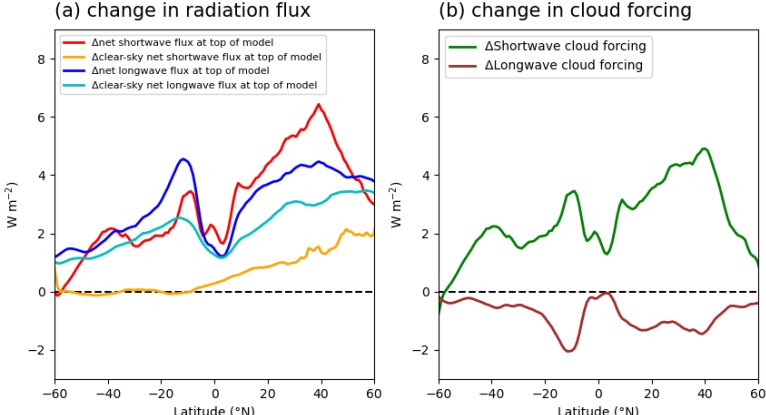

**Figure 3. Zonal averaged change in (a) radiation flux and (b) cloud forcing in W m$^{-2}$ after removal of anthropogenic emissions (Plio_Pristine - Plio_Polluted)**. (a) Change in net shortwave flux at top of model and net longwave flux at top of model under full-sky (red, blue) and clear-sky conditions (orange, cyan). (b) Change in shortwave cloud forcing (green) and longwave cloud forcing (brown). The corresponding spatial change in cloud shortwave and longwave forcings shown in panel b are displayed in Fig. A1a,b respectively.



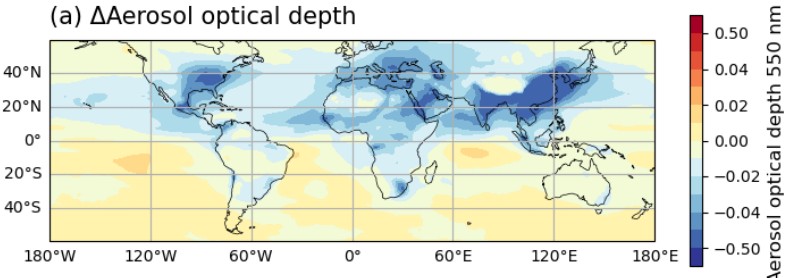

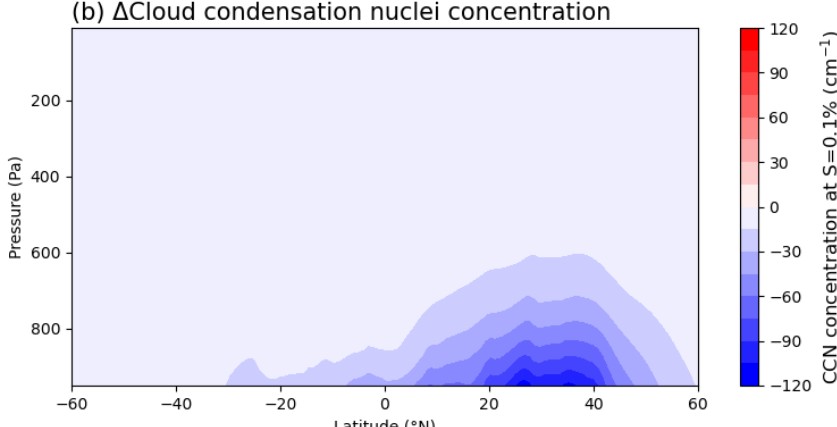

**Figure 4. Change in (a) aerosol optical depth 550 nm and (b) cloud condensation neucli concentration as supersaturation 0.1%**

**(cm$^{-3}$) after removel of anthropogenic emissions (Plio_Pristine - Plio_Polluted).**



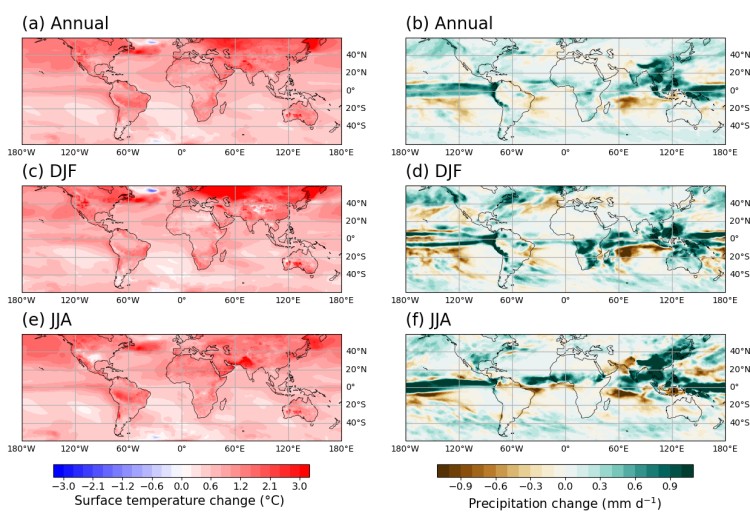

**Figure 5.** Annual, DJF and JJA mean surface temperature change in °C (a, c, e) and precipitation change in mm d$^{-1}$ (b,d,f) after removel of anthropogenic emissions (**Plio_Pristine - Plio_Polluted**).



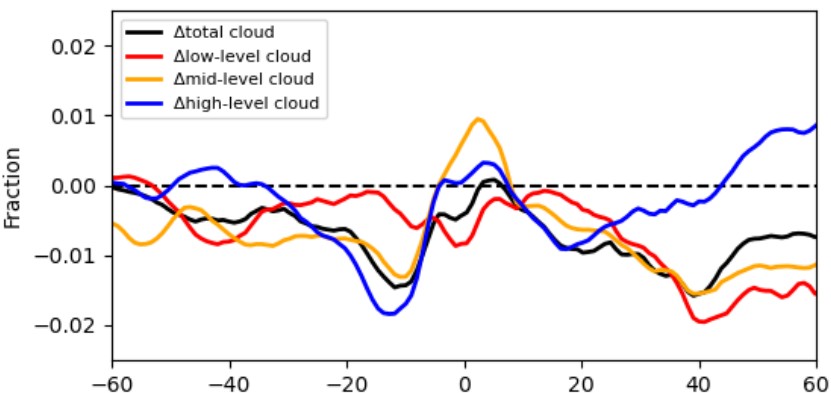

**Figure 6. Zonal averaged change in high-, mid- and low- level clouds.** The corresponding spatial relative change in cloud fractions are given in Fig. A2.





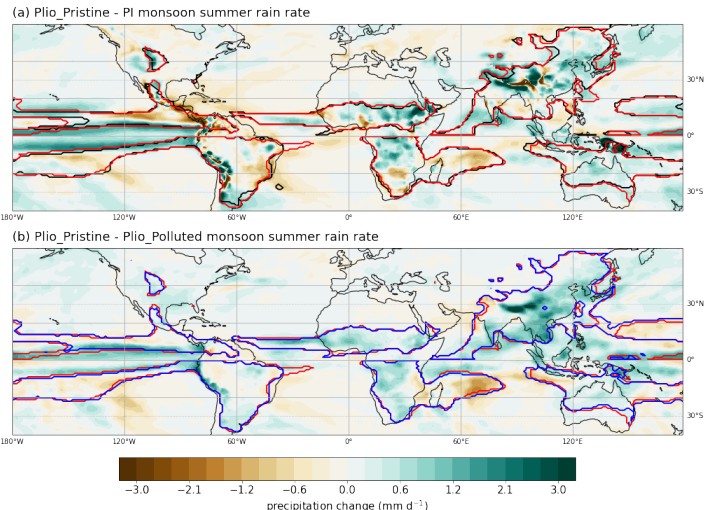

**Figure 7. Changes in monsoon summer rain rate (mm d$^{-1}$)** during the mPWP (Plio_Pristine - PI, panel a) and by removing anthropogenic from the atmosphere (Plio_Pristine - Plio_Polluted, panel b). Black, red and blue contours represent the boundary of the global monsoon domain following Wang et al. (2014) in PI, Plio_Pristine and Plio_Polluted respectively.



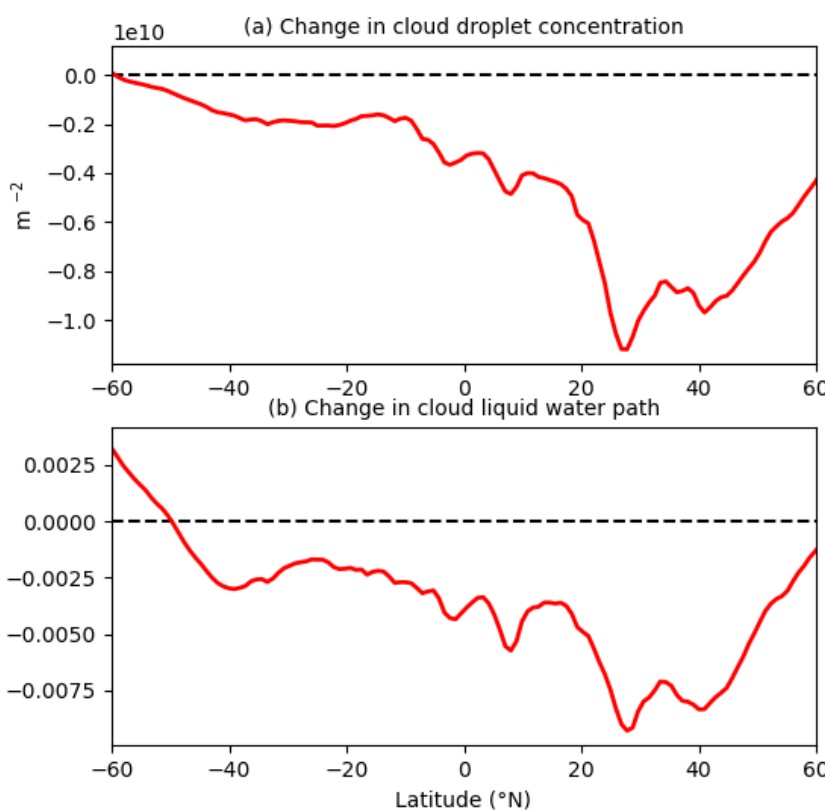

**Figure 8. Zonal averaged change in (a) cloud droplet concentration in m$^{-2}$ and (b) cloud liquid water path in kg m$^{-2}$.** The corresponding spatial relative change are given in Fig. A3.



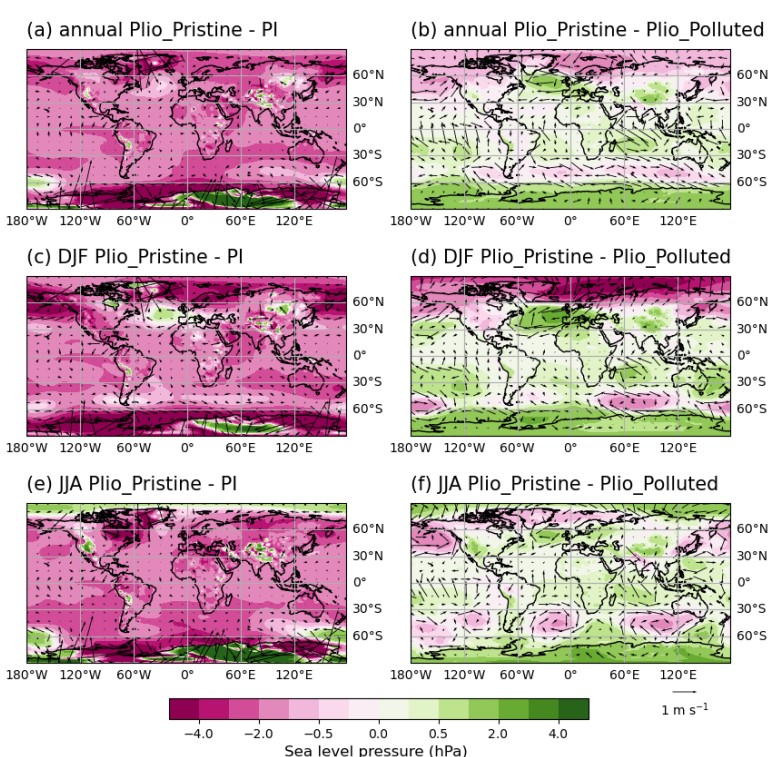

**Figure 9. Changes in annual and seasonal sea level pressure (hPa) and surface wind (m s$^{-1}$)** during the mPWP (Plio_Pristine - PI, left column) and after remocal of anthropogenic aerosols (Plio_Pristine - Plio_Polluted, right column).




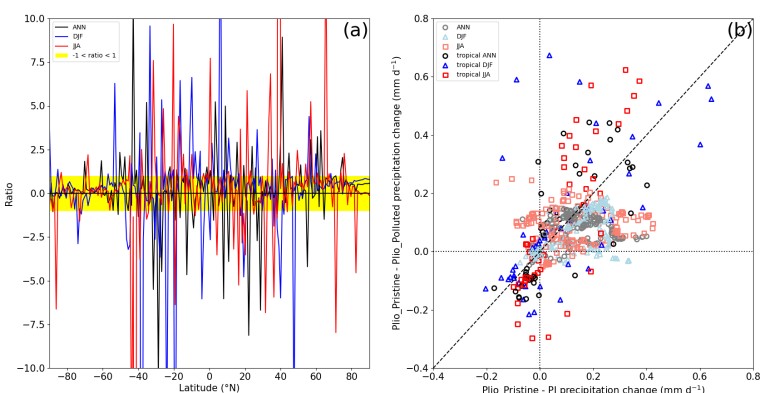

**Figure 10. Relative importance of removing anthropogenic aerosols and mPWP boundary conditions on mPWP zonal mean precipitation change.** (a) Ratio of change in annual, JJA and DJF precipiation change computed as ((Plio_Pristine - Plio_Polluted) / (Plio_Pristine - PI)). Yellow shading highlights where -1< ratio < 1. (b) Comparison between latitude average mean precipitation change induced by PlioMIP boundary conditions and by aerosol scenarios within the tropics (light colors) and outside of the tropics (darker colors).



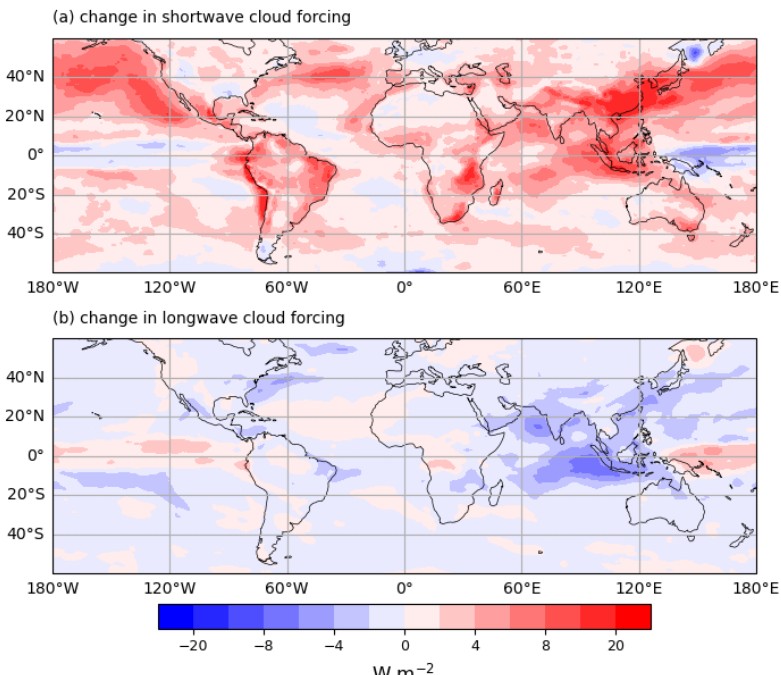

**Figure A1.** Change in (a) shortwave cloud forcing and (b) longwave cloud forcing in W m$^{-2}$ (Plio_Pristine - Plio_Polluted).

**Appendix A:  Cloud properties**



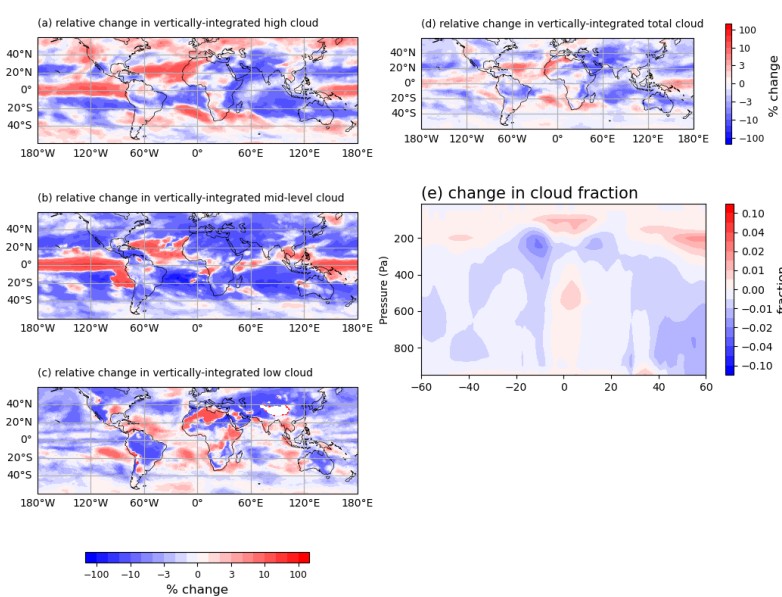

**Figure A2. Spatial relative change in vertically-integrated (a) high, (b) mid-level, (c) low and (d) total cloud ((Plio_Pristine and Plio_Polluted)*100/Plio_Pristine), and vertical change in (e) cloud fraction as difference between Plio_Pristine and Plio_Polluted.**



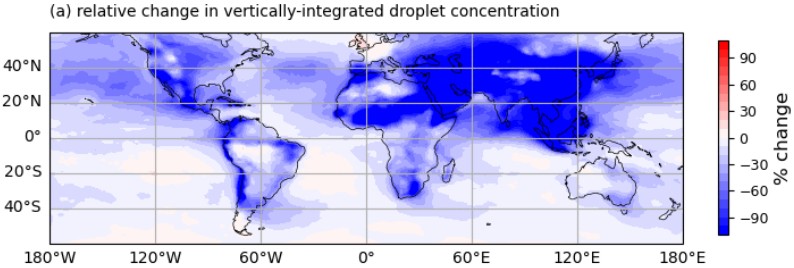

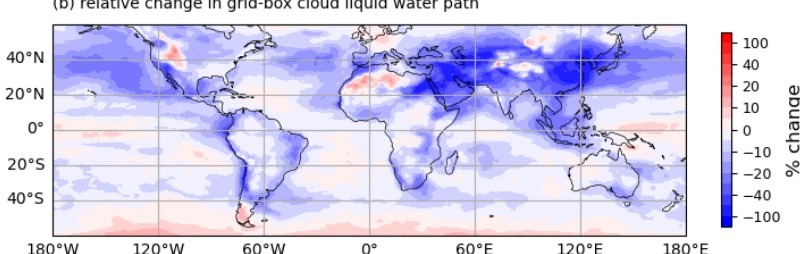

**Figure A3. Spatial relative change in (a) vertically-integrated droplet concentration and (b) grid-box cloud liquid water path as compared to Plio_Pristine (i.e. (Plio_Pristine - Plio_Polluted)*100/Plio_Pristine).**