# Peer review of "Aerosol uncertainties in tropical precipitation changes for the mid-Pliocene Warm Period"

_EGUsphere, 2023_

## Referee Comment (RC2)

**Comments on "Aerosol uncertainties in tropical precipitation changes for the mid-Pliocene Warm Period"**

In this study the authors analysed the climate uncertainties resulted from aerosol scenarios in the simulations of mid-Pliocene Warm Period. Three simulation experiments PI, Plio_Pristine, and Plio_Polluted were analysed, which represent pre-industrial, mid-Pliocene with pre-industrial emissions, and mid-Pliocene with pre-industrial emissions plus industrial pollutants. They found that the change of precipitation between Plio_Pristine and Plio_Polluted is generally larger than that between Plio_Pristine and PI. And they concluded that the aerosol forcing was the dominant driver in tropical precipitation change during mid-Pliocene.

This work seems to be a continuation of the work in Feng et al. (2019) with further analysis of the model results. In my opinion, there are some novel findings but not enough and explicit under current analysis. The details of some results and discussion are missing. And the conclusion is too strong. So I suggest the authors to make a major revision.

The general comments and specific comments are listed below:

**General comments**

1. Please reduce the number of colors used in the colorbars in the figures, it is difficult to relate the colors with the value ranges.

2. With the model experiments designed or shown in this study, it is reasonable to study the aerosol uncertainties with two idealised aerosol scenarios. However, the conclusion (aerosol forcing is the dominant driver in tropical precipitation change in mid-Pliocene) can not be derived following this work. First, in the PlioMIPs, the emissions of aerosols and their precursors were designed to be following the pre-industrial configuration (Haywood et al., 2011), so adding the industrial pollutants is an idealised experiment and not based on proxy data or derived mid-Pliocene conditions. Therefore, you can not say aerosol forcing was the dominant driver in mid-Pliocene, because the forcing was not real. Secondly, you may need more experiments to compare individual components. For example, an experiment with only aerosol change but keeping CO2 in a pre-industrial level, an experiment without aerosol-cloud interaction, etc. You can refer to the experiments in Feng et al. (2019) and Sagoo and Storelvmo (2017).

**Specific comments**

P1, L4: 'is' -> 'it'

P2, L51-53: "mPWP simulations now use modern-day or pre-industrial aerosol concentration that may differ from the conditions during the mPWP. It implies that aerosol effects may be one of the possible explanation for the mismatch between reconstructions and simulations."
The logic is not obvious here between two sentences. Please clarify it or add references.

P3, L62-63: "This increases the shortwave cloud radiative forcing, thus, cools surface temperature and amplifies polar warmth (Sagoo and Storelvmo, 2017)."
Please check the reference, I think if you have cooling effect on the surface, the polar amplification means the amplified polar cooling.

P3, L69-70: "potential analogue nature of Pliocene climate, not to the present-day polluted climate,

but to future climate scenarios which feature removal of anthropogenic pollutants"
Which future scenario do you refer to here? Please specify it.

P3, L77: Write the full name of CCSM4.

P3, L79-82: "which uses Aiken, accumulation and coarse modes to solve number and size concentration of internal condensation and coagulation of different species among modes."
What are "size concentration", "internal condensation"? This sentence is not clear and needs to be rephrased.

P3, L87: Please list key configurations or parameters in CCSM4-PlioMIP1 simulation related to this work.

P3, L87: Does 'direct effect' mean 'aerosol direct effect'? Please clarify it.

P3, Sec. 2.2: Are the experiments the same as the ones in Feng et al. (2019)? From the text in P3, L66-67, it seems that the simulations are the same. If they are the same, did you directly use the output data from their experiments or you rerun the experiments? Please specify it clearly in the text.

P4, L90-92: "The other prescribed pre-industrial emissions plus industrial pollutants of anthropogenic SO2 , sulfate and organic compounds estimated for the 2000s from an gridded (0.5° x 0.5°) emission dataset (hereafter referred as to Plio_Polluted) published in Lamarque et al. (2010)."
The description of the configurations of emissions are not clear. For example:
(1) Does "industrial pollutants of anthropogenic SO2" mean the SO2 emissions from industry sector, excluding all the emissions from other sources like vehicle emissions?
(2) Are other anthropogenic emissions included here, like NOx, black carbon?
(3) Here SO2 could be the precursor of aerosols, and sulfate and organic compounds can be directly emitted or formed from precursors, please provide more details of your emission configurations.

P4, L110: "PI" is used before defined. And please also specify what PI simulation represents, e.g., an average over a period from a specific CMIP5/6 simulation experiment?

P5, L122-123: "The combination regenerates the site sets 35, 72; 22, 77; 79,61; 58,61,79,101 and 65, 66 into new sites located at the center of sets."
Where do the numbers come from? Please clarify it.

P5, L131-133: "The underestimation in Northern Hemisphere warming may be explained by that Plio_Pristine was branched from an earlier CCSM4-PlioMIP1 simulation that had underestimated the warming in the Northern Hemisphere (Rosenbloom et al., 2013)."
What about the results from the latest simulations in PlioMIP2? It would be better to also compare your results with the newer experiment results.

P5, L134-135: "13 out of the 37 sites show a mismatch between simulated surface temperature anomaly and reconstructed SST anomaly smaller than 1.0°C."
Please also add the reference here besides in the figure caption.

P5, L138-139: "The reconstructed zonal temperature gradient along the tropical Pacific Ocean is around 0.5°C greater than simulations (Plio_Pristine - PI)"
How do you define the zonal gradient here, is it the temperature difference between two regions? How the regions are defined?

P5, L147-148: "Precipitation decreases over subtropical oceans, eastern parts of South America and monsoon areas over North America."
The proxy data in Fig. 1b show enhanced precipitation almost the whole globe, with only few sites showing reduced precipitation, which indicates a global scale of the precipitation increasing and a regional/local scale of the decreasing. However, the simulation results show reduced precipitation over a large area, especially the Atlantic Ocean. Could you explain the reason, or could you find any references which mentioned it or tried to explain it?

P6, L155: "Reducing atmospheric aerosols is expected to warm the climate."
It needs more details and references.

P6, L169-171: I do not understand the logic of these two sentences, please clarify it.

P6, L176: Why do you think ITCZ is narrower in your simulation Plio_Pristine compared to Plio_Polluted? Please clarify it.

P6, L183-184: "consistent with the positive correlation between tropical precipitation change and spatial deviations of SST warming from the tropical mean (Xie et al., 2010)"
Then what is the reason of decreasing precipitation over the warming SST areas in Fig. 5?

P7, Sec. 4.1: The discussion in this section about interaction of aerosol and cloud is not clear.
(1) It is better to first give the definitions of net shortwave/longwave flux, clear-sky/full-sky net shortwave/longwave flux, shortwave/longwave cloud forcing, and also how you estimate the aerosol direct effects from these variables. In Fig. 3, the aerosol direct effect is not shown, so you need to clarify it.
(2) "while clouds contribute to a net forcing greater than 2 W m-2 with more shortwave absorption and less longwave emission"
Which latitudes do you mean here? In this phrase "more shortwave absorption and less longwave emission", do you mean on the surface or in the cloud level?
(3) "is still less important than the change in shortwave cloud forcing."
How can you find this from Figs. 3 and 4?
(4) "Figure 6 shows decrease in droplet concentration (especially over lands and high latitudes, Fig. A3a) and decrease in cloud liquid path (except tropical Pacific between 5°S and 5°N and western Africa, Fig. A3b) after the removal of pollutants."
How could you conclude this from Fig. 6?
(5) The title of this section is "Key forcing driving tropical precipitation change", but there is no discussion about how the forcings are driving tropical precipitation change. Please add more discussion details.

P8, L225-228: "Figure 10 shows the relative importance of removing anthropogenic aerosols and mPWP boundary conditions on mPWP zonal mean precipitation change. The ratio (panel a) indicates that the relative importance of effects of aerosol forcing and mPWP boundary conditions (including high CO2) on precipitation is complicated, but overall aerosol effect is more important over the tropics (panel b), which could imply the importance of aerosol scenario in simulating Pliocene climate."
(1) Fig. 10a is difficult to see the features, please find a better way to plot the data, maybe increase the width-height ratio?
(2) Please add more detailed discussion about the relative importance instead of just one word "complicated".
(3) It is difficult to see "aerosol effect is more important over the tropics" from Fig. 10b, maybe a global map of the ratio is better to represent it?

(4) In Figs. 9 and 10 you mentioned the seasonal values and annual values, but you have never discussed any details related to seasonal change. Please add more discussion about it.

P8, L241-242: "SO2 increased by 11 Tg in Unger and Yue (2014) mPWP simulations, yet SO2 from anthropogic sources is estimated to be 120 Tg from 1980 to 2000s (Smith et al., 2011)."
Here 11 Tg should be 11 Tg/yr, which is the emission rate, the same for 120 Tg/yr.

P8, L42-43: "As such, despite that enhanced biogenic emissions may compensate some of the responses seen in the Plio_Pristine"
For "compensate", do you mean: "If the enhanced biogenic emissions, which will increase the secondary organic aerosol formation and growth, is included in the Plio_Pristine simulation, the difference between Plio_Pristine and PI will decrease."? The statement here is not clear, please clarify it.

P8, L246-247: "Sagoo and Storelvmo (2017) looked in to the effect of dust and demonstrated that increased dust emissions resulted in an enhanced dynamical response that lead to great changes in hydrological circulations."
The dust emission in mPWP is lower compared to pre-industrial, why you are talking about "increased dust emissions"?

P8, L247-249: "Their Pliocene simulation with idealised dust scenario shows greater change in precipitation than only consider CO2 forcing, yet only used idealized scenario (Sagoo and Storelvmo, 2017)."
Please add more details, e.g., which scenario, how large is the "greater change in precipitation", global or tropical precipitation.

P9, L251-252: Please add at least some simplified definition of the moisture budget component terms.

P9, L254-255: The statement from "high CO2 concentration" to "thermodynamic change plays an important role" is not explicitly clear here, please add more detailed discussion if you want to keep this statement.

P9, L255-257: Removal of anthropogenic emissions will also affect the thermodynamics itself in the atmosphere, and the anthropogenic emissions include both scattering (e.g., sulfate) and absorbing (e.g., black carbon) compositions, which are also related to shortwave and longwave radiation, so this process is complicated. Therefore, I do not think it "would logically increase the effects of thermodynamic and net energy input". Please add more discussion and references here.

Figs. 6 and 8: Change between which cases? Please specify it in the figure caption.

**References**

Feng, R., Otto-Bliesner, B. L., Xu, Y., Brady, E., Fletcher, T., & Ballantyne, A. (2019). Contributions of aerosol-cloud interactions to mid-Piacenzian seasonally sea ice-free Arctic Ocean. Geophysical Research Letters, 46, 9920–9929. https://doi.org/10.1029/2019GL083960.

Haywood, A. M., Dowsett, H. J., Robinson, M. M., Stoll, D. K., Dolan, A. M., Lunt, D. J., Otto-Bliesner, B., and Chandler, M. A.: Pliocene Model Intercomparison Project (PlioMIP): experimental design and boundary conditions (Experiment 2), Geosci. Model Dev., 4, 571–577, https://doi.org/10.5194/gmd-4-571-2011, 2011.

Sagoo, N., and T. Storelvmo (2017), Testing the sensitivity of past climates to the indirect effects of dust, Geophys. Res. Lett., 44, 5807–5817, doi:10.1002/2017GL072584.

---

## Author Comment (AC2)

**Reply to Referee 3's comments on egusphere-2023-2702**

A. Zhao *et al.*

**Correspondence:** annizhao@pku.edu.cn

The climate in the mid-Pliocene Warm Period (mPWP) is usually viewed as an analog for future climate, but simulated mPWP climate change is largely different from the climate change in the 20th century. Also, there are large discrepancies between the simulation and proxies in the mPWP climate. To address the discrepancies, Zhao et al. suggested a different aerosol scenario from the pre-industrial aerosol setting should be used for simulating the mPWP climate. By removing pollutants of the industrial period in the mPWP climate simulation, the authors get more warming in the Northern Hemisphere and stronger ITCZ. The manuscript is well-written and I suggest a minor revision. My comments are below.

**We would like to thank you for reviewing our manuscript and providing comments about our work. We are happy to adopt many of the comments and will revise the manuscript as requested.**

1. How the results of this study can help resolve the underestimation of temperature increase in the climate models should be addressed. In the abstract and Introduction, the authors mentioned discrepancies between models and proxies in the mPWP. One important difference is that the models underestimate the high temperature in the mPWP, especially in the high latitudes. When simulating the mPWP climate, people often use pre-industrial aerosol forcing, and the results of this study tell us that using pre-industrial aerosol forcing leads to increased warming. Thus, if we use more aerosol forcing, as the authors suggest the mPWP may have more aerosols, the simulated mPWP climate should be even cooler, and the discrepancies between the simulation and proxies would be larger. This confuses me and I think it is important to let readers know how considering aerosols forcing scenario can help improve climate simulations in the mPWP.

**Thanks for raising the question, but there might be a misunderstanding here. Our results show opposite finding actually. Results of this study do not mention that using pre-industrial aerosol forcing leads to increased warming. Sect. 3.1 describes the warming driven by prescribed mPWP boundary conditions (Plio_Pristine - PI) and compares the simulated mPWP change with the reconstructions, which points out the existence of data-model mismatch. The mPWP climate is expected to have less aerosols than pre-industrial. Sect. 3.2 and 3.3 analyse the effect of removing anthropogenic aerosol emissions (Plio_Pristine - Plio_Polluted), which actually show that less aerosols lead to further warming. It means that if we use less aerosols, the simulated mPWP climate should be furhter warm, and the discrepancies between the simulation and proxies would be smaller.**

2. The title "Aerosol uncertainties in tropical precipitation changes for the mid-Pliocene Warm Period" confuses me. What does "aerosol uncertainties in tropical precipitation" mean? Does it mean the uncertainties of aerosol bring uncertainties in tropical precipitation in climate models?

**"aerosol uncertainties in tropical precipitation" means the uncertainties induced by aerosol forcing in simulating (mPWP) climate.**

L28: Repeated "e.g."s

**We will delete the extra "e.g.".**

L39: "emission induced": What emission? GHGs or aerosols?

**It means aerosol emission. We will add "aerosol" in text.**

L118 and L121: (Feng et al., 2022) -> Feng et al. (2022)

**We will fix it.**

Figure 1: What are the circles in Fig. 1b? Do the proxies in Fig. 1b represent precipitation minus evaporation? If so, why not compare them to the simulated P-E? Also, please increase the white space between Figure 1a and 1b.

**Circles in Fig.1b are reconstructed annual mean precipitation change published in Salzmann et al. (2008). Triangles are signals of hydroclimate (precipitation minus evaporation) change published in Feng et al. (2022). The aim of panel b is to show precipitation change rather than hydroclimate change. It is difficult to have abundant quantitative evidence of mPWP precipitation anomaly. Therefore, we choose to plot this new qualitative wet-dry dataset to give an idea of the sign of precipitation change during the mPWP, though the compilation does not show precipitation anomaly directly. We will enlarge space between panels.**

L169-171: If "removing aerosols causing more warming than the mPWP boundary conditions", then why do you say that "the changes in boundary conditions are more important than aerosols?" This seems to contradict each other.

**We are trying to say that removing aerosols only causes more over Northeastern Pacific and high-latitude North Atlantic Ocean than the mPWP boundaries. We will rewrite the sentence to "However, the mPWP boundary conditions causes more warming than removing anthropogenic aerosols nearly global except Northeastern Pacific and high-latitude North Atlantic Ocean (Fig. 5a,c,e). More warming lead by the mPWP boundaries than removal of aerosols suggests ..." to emphasise that the mPWP boundaries lead to more warming than removal of aerosols.**

L212: Figure 6 is about droplet concentration. This should be Figure 7.

**There was extra panels about annual averaged change in droplet concentration and cloud liquid path in our old Fig.6. We feel Fig A3 would be sufficient, so we removed these panels before submission. We do apologise. We will rewrite this sentence to "Removal of pollutants decreases droplet concentration (especially over lands and high latitudes, Fig. A3a) and decrease cloud liquid path (except tropical Pacific between 5°S and 5°N and western Africa, Fig. A3b)."**

65    L219: "general increase in sea level pressure": But the left panel of Figure 9 shows an increase in sea level pressure.

**Sorry we made a mistake here. We will rewrite the sentence to "Over tropical and subtropical regions, the overall uniform warming induce by mPWP boundaries (Plio_Pristine - PI) result in general decrease in sea level pressure and have relatively small effects on surface wind (Fig. 9a,c,e). Though removal of anthropogenic aerosols also show warming in tropical and subtropical regions, it rises the sea level pressure and have much stronger effects on surface wind that**

70    **shows seasonal variances (Fig. 9b,d,f)."**

L227-228: I cannot see the "overall aerosol effect is more important" from Figure 10b. The numbers of dots above and below the 1:1 line are nearly equal.

**We will replot Fig.10.**

75

L270: due to mPWP had warmer and wetter climate -> due to warmer and wetter climate in the mPWP

**We will revise it.**

**References**

80    Feng, R., Bhattacharya, T., Otto-Bliesner, B. L., Brady, E. C., Haywood, A. M., Tindall, J. C., Hunter, S. J., Abe-Ouchi, A., Chan, W.-L., Kageyama, M., Contoux, C., Guo, C., Li, X., Lohmann, G., Stepanek, C., Tan, N., Zhang, Q., Zhang, Z., Han, Z., Williams, C. J. R., Lunt, D. J., Dowsett, H. J., Chandan, D., and Peltier, W. R.: Past terrestrial hydroclimate sensitivity controlled by Earth system feedbacks, Nature Communications, 13, 1306, https://doi.org/10.1038/s41467-022-28814-7, 2022.

Salzmann, U., Haywood, A. M., Lunt, D. J., Valdes, P. J., and Hill, D. J.: A new global biome reconstruction and data-model comparison for

85    the Middle Pliocene, Global Ecology and Biogeography, 17, 432–447, https://doi.org/10.1111/j.1466-8238.2008.00381.x, 2008.

---

## Author Comment (AC3)

**Reply to Referee 2's comments on egusphere-2023-2702**

A. Zhao *et al.*

**Correspondence:** annizhao@pku.edu.cn

Comments on "Aerosol uncertainties in tropical precipitation changes for the mid-Pliocene Warm Period"

In this study the authors analysed the climate uncertainties resulted from aerosol scenarios in the simulations of mid-Pliocene Warm Period. Three simulation experiments PI, Plio_Pristine, and Plio_Polluted were analysed, which represent pre-industrial, mid-Pliocene with pre-industrial emissions, and mid-Pliocene with pre-industrial emissions plus industrial pollutants. They found that the change of precipitation between Plio_Pristine and Plio_Polluted is generally larger than that between Plio_Pristine and PI. And they concluded that the aerosol forcing was the dominant driver in tropical precipitation change during mid-Pliocene.

This work seems to be a continuation of the work in Feng et al. (2019) with further analysis of the model results. In my opinion, there are some novel findings but not enough and explicit under current analysis. The details of some results and discussion are missing. And the conclusion is too strong. So I suggest the authors to make a major revision.

The general comments and specific comments are listed below:

**We would like to thank the reviewer for commenting our manuscript. We are happy to adopt many of the comments and will revise the manuscript as requested.**

General comments

1. Please reduce the number of colors used in the colorbars in the figures, it is difficult to relate the colors with the value ranges.

**Figures will be replotted.**

2. With the model experiments designed or shown in this study, it is reasonable to study the aerosol uncertainties with two idealised aerosol scenarios. However, the conclusion (aerosol forcing is the dominant driver in tropical precipitation change in mid-Pliocene) can not be derived following this work. First, in the PlioMIPs, the emissions of aerosols and their precursors were designed to be following the pre-industrial configuration (Haywood et al., 2011), so adding the industrial pollutants is an idealised experiment and not based on proxy data or derived mid-Pliocene conditions. Therefore, you can not say aerosol forcing was the dominant driver in mid-Pliocene, because the forcing was not real. Secondly, you may need more experiments to compare individual components. For example, an experiment with only aerosol change but keeping $CO_2$ in a pre-industrial level, an experiment without aerosol-cloud interaction, etc. You can refer to the experiments in Feng et al. (2019) and Sagoo and Storelvmo (2017).

**We appreciate that the aerosol forcing that we investigated are idealised, and therefore cannot hope to quantify the**

30    **amount of uncertainty in Pliocene aerosol. However, we have demonstrated that it is could as large that resulting from other boundary conditions, in certain variables such as tropical rainfall. We had aimed to convey aerosols can be the dominant source of uncertainty, rather than the dominant driver. It is clear from the reviewers' comments that we have not made that as clear as we could have. In a revised manuscript, we will rephrase the conclusions to draw out this distinction.**

35    **We are not aware of any research that aerosol forcing and the resulted climate responses are state-dependent to the first order. This means that comparing Plio_Pristine with Plio_Polluted should yield the same results to comparing PI_Pristine and PI_Polluted (which we found during our initial investigations). For the latter, an experiment without ACI would be CCSM4 experiments with Plio-Pristine and Plio-Polluted. However, we don't have the computing resource to run those simulations. We want to point out that the direct aerosol forcing without ACI is very small (Myhre**

40    **et al., 2013). Having those simulations would not change our results and conclusion.**

Specific comments

P1, L4: 'is' -> 'it'

**Will be fixed.**

45

P2, L51-53: "mPWP simulations now use modern-day or pre-industrial aerosol concentration that may differ from the conditions during the mPWP. It implies that aerosol effects may be one of the possible explanation for the mismatch between reconstructions and simulations." The logic is not obvious here between two sentences. Please clarify it or add references.

**The existed data-model mismatch during the mPWP implies that the mPWP simulations haven't been able to include**

50    **all the mechanisms nor to use realistic prescribed inputs. The highlight sentences were trying to point out that the mPWP simulations might be improved if use more realistic aerosol, as the simulations usually use prescribed aerosol concentration same as the pre-industrial control run. We will rewrite it to "The discrepancy between mPWP simulations and reconstructions (Haywood et al., 2013, 2020) implies that models might miss some important mechanisms (Fedorov et al., 2013) or prescribed forcing could be a source of uncertainty (Feng et al., 2019). The mPWP simulations now use**

55    **modern-day or pre-industrial aerosol concentration same as the control runs that may differ from the conditions during mPWP. Lack of including the effect of aerosol forcings and the usage of unrealistic prescribed aerosol concentration in mPWP simulations implies that aerosol effects may be one of the possible explanation for the mismatch between reconstructions and simulations."**

60    P3, L62-63: "This increases the shortwave cloud radiative forcing, thus, cools surface temperature and amplifies polar warmth (Sagoo and Storelvmo, 2017)." Please check the reference, I think if you have cooling effect on the surface, the polar amplification means the amplified polar cooling.

**Yes, you are right. Simulations of Sagoo and Storelvmo (2017) suggest high dust result in surface temperature cooling and polar amplification (from their high dust and LGM simulations) and low dust result in surface temperature warm-ing and polar amplification (from their low dust and mPWP simulations. We will rewrite this part to only use the result of their fully coupled low dust simulations (a low dust run and a mPWP run modified by low dust). "Results showed that increased dust reduces the size of ice crystals in clouds while increasing their amount. This increases the shortwave cloud radiative forcing, thus, cools surface temperature and amplifies polar warmth (Sagoo and Storelvmo, 2017)." will be revised to "Their mPWP simulation modified by extreme low dust suggests surface temperature warming and polar amplification due to reduced radiative forcing by increased size while decreased amount of ice crystals in clouds."**

P3, L69-70: "potential analogue nature of Pliocene climate, not to the present-day polluted climate, but to future climate scenarios which feature removal of anthropogenic pollutants" Which future scenario do you refer to here? Please specify it.

**Most scenarios for future projections in the IPCC AR6 have a decrease in the anthropogenic aerosol forcing, which will contribute to a increase in global mean annual surface temperature and precipitation (Lee et al., 2021). We will add a reference in text.**

P3, L77: Write the full name of CCSM4.

**We will add it's full name the Community Climate System Model version 4 (CCSM4, Gent et al., 2011) in text.**

P3, L79-82: "which uses Aiken, accumulation and coarse modes to solve number and size concentration of internal con-densation and coagulation of different species among modes." What are "size concentration", "internal condensation"? This sentence is not clear and needs to be rephrased.

**That should be size and number concentration instead. MAM3 solves for sizes and number concentrations of internal mixtures of dust, sea-salt, sulfate, organics and black carbon using three categorical size bins of 0.02–0.08 $\mu$m (Aiken mode), 0.08–1 $\mu$m (Accumulation mode) and 1.0 – 10 $\mu$m (Coarse mode). We will rewrite the sentence to "which uses Aiken (0.02–0.08 $\mu$m), accumulation (0.08–1 $\mu$m) and coarse (1.0–10 $\mu$m) modes to solve size and number concentration of internal condensation and coagulation of different species among modes."**

P3, L87: Please list key configurations or parameters in CCSM4-PlioMIP1 simulation related to this work.

**The CCSM4-PlioMIP1 simulation was generated by CCSM4, an earlier version of CESM. Key improvements be-tween the two generations have been described Meehl et al. (2013) in as well as in Sect. 2.1. Prescribed mPWP bound-ary condition should have no effect on simulations, as the CCSM4-PlioMIP1, Plio_Pristine and Plio_Polluted all used PlioMIP1 boundary condition (Haywood et al., 2010; Dowsett, 2007). Plio_Pristine and Plio_Polluted were branched from the CCSM4-PlioMIP1 simulation, because it would save time and computing cost of letting model reach its equi-librium state. CESM would only need to response to change in aerosol forcing. Therefore, we feel there is no need to list**

**key configurations or parameters of CCSM4-PlioMIP1 simulation.**

P3, L87: Does 'direct effect' mean 'aerosol direct effect'? Please clarify it.

**Yes. "aerosol" will be added.**

P3, Sec. 2.2: Are the experiments the same as the ones in Feng et al. (2019)? From the text in P3, L66-67, it seems that the simulations are the same. If they are the same, did you directly use the output data from their experiments or you rerun the experiments? Please specify it clearly in the text.

**Yes, simulations are the same, which is mentioned in Line 66-67 "The simulations were performed initially to analyse ..." We will rewrite the sentences to "In this study, we analyse two Pliocene simulations that were performed initially to analyse the effects of aerosol-cloud interactions on mPWP seasonally sea ice-free Arctic (Feng et al., 2019) to further investigate the potential effect of aerosol on mPWP climate. Different aerosol scenarios were applied, one with pre-industrial aerosol concentrations and one with present-day aerosol concentrations published in Lamarque et al. (2010)." We will also emphasise it again in Sect. 2.2 by adding "The existed simulations (Feng et al., 2019)..." at the beginning.**

P4, L90-92: "The other prescribed pre-industrial emissions plus industrial pollutants of anthropogenic SO2 , sulfate and organic compounds estimated for the 2000s from an gridded (0.5° x 0.5°) emission dataset (hereafter referred as to Plio_Polluted) published in Lamarque et al. (2010)." The description of the configurations of emissions are not clear. For example: (1) Does "industrial pollutants of anthropogenic SO2" mean the SO2 emissions from industry sector, excluding all the emissions from other sources like vehicle emissions? (2) Are other anthropogenic emissions included here, like NOx, black carbon? (3) Here SO2 could be the precursor of aerosols, and sulfate and organic compounds can be directly emitted or formed from precursors, please provide more details of your emission configurations.

**Lamarque et al. (2010) published a dataset of historical (1850-2000) anthropogenic (defined as originating from industrial, domestic and agriculture activity sectors) and biomass burning emissions of reactive gases and aerosols. The dataset provided consistent gridded emissions for CMIP5 models to run chemistry model simulations that contributed to the assessment in IPCC AR5. The emissions for year 2000 of the dataset, which prescribed in Plio_Polluted, serve as an anchor point for historical emissions and future emissions Lamarque et al. (2010). (1) "industrial" is relative to "pre-industrial" ahead, which refers to time instead of a source. (2) No, NOx and black carbon were excluded. Emissions were constant in the Plio_Pristine and Plio_Polluted. (3) Lamarque et al. (2010) is an emission dataset, i.e. from primary source. Secondary sources are not included. Emitting configuration is well documented in Lamarque et al. (2010) as cited in text. We will add a bit more to introduce the dataset by "... published in Lamarque et al. (2010). The emission dataset provided consistent gridded anthropogenic (defined as originating from industrial, domestic and agriculture activity sectors) and biomass burning of reactive gases and aerosols covering the historical period from 1850 to 2000 CMIP5 models to use in running chemistry model simulations that would contributed to the assessment in**

**IPCC AR5 (Lamarque et al., 2010)."**

P4, L110: "PI" is used before defined. And please also specify what PI simulation represents, e.g., an average over a period from a specific CMIP5/6 simulation experiment?

**PI will be defined here. In modelling, PI usually refers to the pre-industrial control run, which applies pre-industrial (1850 CE) boundary condition and performs as a baseline for computing anomaly. In this study, PI refers to a simulation run by CESM model with pre-industrial boundary condition prescribed. We think there is no need to specify it in text.**

P5, L122-123: "The combination regenerates the site sets 35, 72; 22, 77; 79,61; 58,61,79,101 and 65, 66 into new sites located at the center of sets." Where do the numbers come from? Please clarify it.

**Site numbers come from the the compilation of mPWP proxy records of Feng et al. (2022). We will add its source in text, as "The combination regenerates the site sets 35, 72; 22, 77; 79,61; 58,61,79,101 and 65, 66 of Feng et al. (2022) into new sites located at the center of sets."**

P5, L131-133: "The underestimation in Northern Hemisphere warming may be explained by that Plio_Pristine was branched from an earlier CCSM4-PlioMIP1 simulation that had underestimated the warming in the Northern Hemisphere (Rosenbloom et al., 2013)." What about the results from the latest simulations in PlioMIP2? It would be better to also compare your results with the newer experiment results.

**First of all, the simulations used PlioMIP1 boundary conditions. Secondly, though PlioMIP2 simulations produce warmer and wetter mPWP than PlioMIP1, the difference is caused by the addition of new and more sensitive models instead of the modifications in boundary conditions between the two PlioMIPs (Haywood et al., 2020). CESM's earlier version CCSM4 participated in both PlioMIPs and show that PlioMIP2 anomalies are similar to PlioMIP1's (Haywood et al., 2020). Therefore, it would be appropriate to compare with the PlioMIP1 outputs here. Comparsion with the PlioMIP2 results would not be necessary.**

P5, L134-135: "13 out of the 37 sites show a mismatch between simulated surface temperature anomaly and reconstructed SST anomaly smaller than 1.0°C." Please also add the reference here besides in the figure caption.

**Foley and Dowsett (2019) will be added.**

P5, L138-139: "The reconstructed zonal temperature gradient along the tropical Pacific Ocean is around 0.5°C greater than simulations (Plio_Pristine - PI)" How do you define the zonal gradient here, is it the temperature difference between two regions? How the regions are defined?

**The reconstructed temperature gradient is calculated as the temperature difference between sites near Colombia and sites near SE Asia. We will give the regions and change "zonal" to "meridional" for being easier to understand.**

P5, L147-148: "Precipitation decreases over subtropical oceans, eastern parts of South America and monsoon areas over

North America." The proxy data in Fig. 1b show enhanced precipitation almost the whole globe, with only few sites showing reduced precipitation, which indicates a global scale of the precipitation increasing and a regional/local scale of the decreasing. However, the simulation results show reduced precipitation over a large area, especially the Atlantic Ocean. Could you explain the reason, or could you find any references which mentioned it or tried to explain it?

170 **The spatial pattern of the increased precipitation over low and high latitudes and the reduction over part of the parts of the subtropics in Fig.1b is consistent with the PlioMIPs results (Haywood et al., 2013, 2020). In the mPWP simulations, precipitation occurs more in the tropics than the subtropics by having a weakened Hadley circulation that is related to decreased zonal temperature gradient between the tropics and the subtropics (Corvec and Fletcher, 2017). Models also produce a poleward shift in mid-latitude westerlies during the mPWP, which is linked to reduced merid-**

175 **ional temperature gradient (Li et al., 2015). The poleward shift in mid-latitude westerlies results in a poleward shift in higher-latitude precipitation. The precipitation over subtropics, thus, shows a reduction in the mPWP simulations.**

P6, L155: "Reducing atmospheric aerosols is expected to warm the climate." It needs more details and references.

**Aerosols that mainly scatter solar radiation have a cooling effect, and those absorb solar radiation have a cooling effect. Scattering and absorbing aerosols mix in the atmosphere. Net effect of aerosols on energy budget is dependent**

180 **on the characteristics of surface and cloud. According to IPCC AR6, human-induced aerosols contributed to a cooling of 0.0°C to 0.8°C between 2010-2019 relative to 1850-1900 (IPCC, 2021). Therefore, removing anthropogenic aerosols is expected to warm the climate. We will add the later sentence (According to IPCC AR6 ... (IPCC, 2021).) in text.**

P6, L169-171: I do not understand the logic of these two sentences, please clarify it.

185 **We are trying to say that removing aerosols only causes more over Northeastern Pacific and high-latitude North Atlantic Ocean than the mPWP boundaries. We will rewrite the sentence to "However, the mPWP boundary conditions causes more warming than removing anthropogenic aerosols nearly global except Northeastern Pacific and high-latitude North Atlantic Ocean (Fig. 5a,c,e). More warming lead by the mPWP boundaries than removal of aerosols suggests ..." to emphasise that the mPWP boundaries lead to more warming than removal of aerosols.**

190

P6, L176: Why do you think ITCZ is narrower in your simulation Plio_Pristine compared to Plio_Polluted? Please clarify it.
**We will discussion on change in Hadley Circulation.**

P6, L183-184: "consistent with the positive correlation between tropical precipitation change and spatial deviations of SST

195 warming from the tropical mean (Xie et al., 2010)" Then what is the reason of decreasing precipitation over the warming SST areas in Fig. 5?

**We focus on annual change here, which shows an overall enhancement along the tropics that is largely affected by the strong increase during JJA. If you mean the decreasing precipitation over the tropics during DJF, a possible explanation is due to the change in low-latitude easterlies (Fig.9d), as the low-latitude easterlies is strengthened between 30°N(S)**

200    **and roughly 10°N(S) while is weakened between 10°N(S) over the tropical Pacific.**

P7, Sec. 4.1: The discussion in this section about interaction of aerosol and cloud is not clear. (1) It is better to first give the definitions of net shortwave/longwave flux, clear-sky/full-sky net shortwave/longwave flux, shortwave/longwave cloud forcing, and also how you estimate the aerosol direct effects from these variables. In Fig. 3, the aerosol direct effect is not shown, so

205    you need to clarify it.

   **Fig.3a plots change in net shortwave and longwave flux at top of model under full-sky and clear-sky condition, respectively. At top of model can be viewed as at top of the atmosphere (TOA). Net shortwave/longwave flux are TOA shortwave absorption and longwave emission. clear-sky/full-sky defines the cloud status, i.e. full-sky means cloudy sky and clear-sky refers to no clouds. For example, clear-sky net shortwave flux means TOA shortwave absorption with no**

210    **clouds in the atmosphere. Clear-sky variables show aerosol direct effect. Difference between net full-sky (cloudy sky) and clear-sky (no clouds) flux would show aerosol indirect effect, which can be understood as cloud forcing. Shortwave cloud forcing is measured by difference in surface downwelling shortwave radiation between clear-sky condition and the full condition, while longwave forcing is the difference in TOA upwelling longwave radiation.**

215    (2) "while clouds contribute to a net forcing greater than 2 W m-2 with more shortwave absorption and less longwave emission" Which latitudes do you mean here? In this phrase "more shortwave absorption and less longwave emission", do you mean on the surface or in the cloud level?

   **Sect. 4.1 focuses on tropical precipitation change. The "2 W m-2" comes from tropical average (23.5°N to 23.5°S), which can be seen in Fig.3b. "more shortwave absorption and less longwave emission" refers to net forcing change**

220    **induced by clouds, which are surface shortwave cloud forcing and and TOA longwave cloud forcing.**

   (3) "is still less important than the change in shortwave cloud forcing." How can you find this from Figs. 3 and 4?

   **Aerosol affect energy flux through direct effect by reflection or indirect effect through interaction with clouds.Difference between net full-sky (cloudy sky) and clear-sky (no clouds) flux would show aerosol indirect effect. Shortwave cloud forc-**

225    **ing is measured by subtracting surface downwelling shortwave radiation of clear-sky condition from the full condition. The difference patterns (not shown in figures directly) match the change in cloud forcing, which shows that aerosol indirect effect is more important here.**

   (4) "Figure 6 shows decrease in droplet concentration (especially over lands and high latitudes, Fig. A3a) and decrease in

230    cloud liquid path (except tropical Pacific between 5°S and 5°N and western Africa, Fig. A3b) after the removal of pollutants." How could you conclude this from Fig. 6?

   **We do apologise. There was extra panels about annual averaged change in droplet concentration and cloud liquid path in our old Fig.6. We feel Fig A3 would be sufficient, so we removed these panels before submission. We will rewrite this sentence to "Removal of pollutants decreases droplet concentration (especially over lands and high latitudes, Fig.**

**A3a) and decrease cloud liquid path (except tropical Pacific between 5°S and 5°N and western Africa, Fig. A3b)."**

(5) The title of this section is "Key forcing driving tropical precipitation change", but there is no discussion about how the forcings are driving tropical precipitation change. Please add more discussion details.

**Analysis in Sect. 3 shows that tropical precipitation enhancement after the removal of anthropogenic emissions is due to the further warming by removing aerosol-induced cooling from the atmosphere. The aim of this section is finding out which aerosol effect (direct/indirect) is more important. The first half discussion shows that aerosol radiation forcing is less important than cloud forcing, i.e. aerosol indirect effect is more important. The second half shows that removing anthropogenic emissions affects cloud forcing by reducing cloud albedo and decreasing cloud lifetime.**

P8, L225-228: "Figure 10 shows the relative importance of removing anthropogenic aerosols and mPWP boundary conditions on mPWP zonal mean precipitation change. The ratio (panel a) indicates that the relative importance of effects of aerosol forcing and mPWP boundary conditions (including high CO2) on precipitation is complicated, but overall aerosol effect is more important over the tropics (panel b), which could imply the importance of aerosol scenario in simulating Pliocene climate." (1) Fig. 10a is difficult to see the features, please find a better way to plot the data, maybe increase the width-height ratio?

**We will replot Fig.10**

(2) Please add more detailed discussion about the relative importance instead of just one word "complicated".

**"complicated" comes form the the ratio of change in latitudial averaged annual, JJA and DJF precipitation change shown in Fig.10a whose patterns are complex. We are trying to detect which (aerosol forcing or mPWP boundary condition) is more important but not why. Overall, Plio_Pristine - Plio_Polluted shows greater change in tropical zonal averaged annual mean precipitation than Plio_Pristine - PI, as shown in Fig. 1 below.**

(3) It is difficult to see "aerosol effect is more important over the tropics" from Fig. 10b, maybe a global map of the ratio is better to represent it?

**Before hand, we apologise that we made a mistake in caption, as dark markers actually show latitudal-averaged change within the tropics and the light colors for those outside of tropics. We'll replot Fig. 10.**

(4) In Figs. 9 and 10 you mentioned the seasonal values and annual values, but you have never discussed any details related to seasonal change. Please add more discussion about it.

**Our analysis is focusing on annual change. The reason of giving seasonal change is to show if the annual change is largely affected by seasonal variance.**

[Figure]

**Figure 1. Latitude-averaged annual mean precipitation change between experiments.**

P8, L241-242: "SO2 increased by 11 Tg in Unger and Yue (2014) mPWP simulations, yet SO2 from anthropogic sources is estimated to be 120 Tg from 1980 to 2000s (Smith et al., 2011)." Here 11 Tg should be 11 Tg/yr, which is the emission rate, the same for 120 Tg/yr.

**We will correct the unit.**

P8, L42-43: "As such, despite that enhanced biogenic emissions may compensate some of the responses seen in the Plio_Pristine" For "compensate", do you mean: "If the enhanced biogenic emissions, which will increase the secondary organic aerosol formation and growth, is included in the Plio_Pristine simulation, the difference between Plio_Pristine and PI will decrease."? The statement here is not clear, please clarify it.

**Sect. 4.3 is for implications for simulating Pliocene climate, not only focusing on the results of this study. According to sentences before Line242-243, though mPWP was expected to have stronger biogenic emissions from more active wildfire and biome productions, the mPWP emission is likely much smaller than the anthropogenic emission during the historical period (1850CE onwards till now) and near future. "compensate" here means the reduced difference between mPWP and near future climate. Because the compensation is small, we make an argument in the following half sentence as "we argue that the absence of anthropogenic pollutant from the mPWP troposphere remains one of the important factors driving the differences between the mPWP and near future climate, especially the Arctic climate (Feng et al., 2019) and hydroclimate."**

P8, L246-247: "Sagoo and Storelvmo (2017) looked in to the effect of dust and demonstrated that increased dust emissions resulted in an enhanced dynamical response that lead to great changes in hydrological circulations." The dust emission in mPWP is lower compared to pre-industrial, why you are talking about "increased dust emissions"?

290 **This sentence is about Sagoo and Storelvmo (2017)'s major finding based on their high-dust simulations, rather than their mPWP simulation. Their mPWP simulation used low-dust scanario. We will remove this sentence from text, and only include the finding of the mPWP simulation.**

P8, L247-249: "Their Pliocene simulation with idealised dust scenario shows greater change in precipitation than only
295 consider CO2 forcing, yet only used idealized scenario (Sagoo and Storelvmo, 2017)." Please add more details, e.g., which scenario, how large is the "greater change in precipitation", global or tropical precipitation.

**We will rewrite the sentence to "Their Pliocene simulation with idealised extreme low dust scenario shows greater change in precipitation than only consider CO2 forcing, yet only used idealized scenario (Sagoo and Storelvmo, 2017)".**

300 P9, L251-252: Please add at least some simplified definition of the moisture budget component terms.

**We will add definitions here.**

P9, L254-255: The statement from "high CO2 concentration" to "thermodynamic change plays an important role" is not explicitly clear here, please add more detailed discussion if you want to keep this statement.

305 **L251-258 discussed which decomposed component of the anomalous moisture budget majorly drives the monsoon change in simulations forced by high-CO2 concentration. The decomposed components include thermodynamic component, dynamic component, residual component and net energy input component. We will revise the text to make it clear.**

310 P9, L255-257: Removal of anthropogenic emissions will also affect the thermodynamics itself in the atmosphere, and the anthropogenic emissions include both scattering (e.g., sulfate) and absorbing (e.g., black carbon) compositions, which are also related to shortwave and longwave radiation, so this process is complicated. Therefore, I do not think it "would logically increase the effects of thermodynamic and net energy input". Please add more discussion and references here.

**We agree that the process is complicated. However, black carbon is not included in our simulations. And as mentioned**
315 **above, "thermodynamic and net energy input" here are the decomposed component of the anomalous moisture budget driving monsoon change.**

Figs. 6 and 8: Change between which cases? Please specify it in the figure caption.

**We will and "Plio_Pristine - Plio_Polluted" in captions to clarify the difference plotted in figures.**

320

References Feng, R., Otto-Bliesner, B. L., Xu, Y., Brady, E., Fletcher, T., & Ballantyne, A. (2019). Contributions of aerosol-cloud interactions to mid-Piacenzian seasonally sea ice-free Arctic Ocean. Geophysical Research Letters, 46, 9920–9929. https://doi.org/10.1029/2019GL083960. Haywood, A. M., Dowsett, H. J., Robinson, M. M., Stoll, D. K., Dolan, A. M., Lunt,

D. J., Otto- Bliesner, B., and Chandler, M. A.: Pliocene Model Intercomparison Project (PlioMIP): experimental design and boundary conditions (Experiment 2), Geosci. Model Dev., 4, 571–577, https://doi.org/10.5194/gmd-4-571-2011, 2011.

Sagoo, N., and T. Storelvmo (2017), Testing the sensitivity of past climates to the indirect effects of dust, Geophys. Res. Lett., 44, 5807–5817, doi:10.1002/2017GL072584.

---

## Author Response (AR2)

**Reply to the referees' comments on egusphere-2023-2702**

A. Zhao *et al.*

**Correspondence:** annizhao@pku.edu.cn

**Summary of changes**

We would like to thank the referees for reviewing our manuscript and providing their kind comments about our work. We are happy and have adopted many of the comments and revised the manuscript as requested. As two of the referees commented on that our manuscript had some scientific uncertainties and missed the details of some results and discussion, we have revised the results, discussion and conclusion sections to clarify and emphasise our findings. We have renamed the Sect. 4.1 "Key forcing driving tropical precipitation change" to "Key forcing contributing to aerosol effect" and revised text to demonstrate that aerosol forcing could be a major source of uncertainty in simulating Pliocene climate change, since it could contribute changes as large as that resulting from other boundary conditions in certain variables such as tropical rainfall. We have made the improvements to the figures as requested. An extra figure showing change in vertical velocity and zonal mean stream function has been added as the new Fig.7 to show change in the ITCZ after removal of anthropogenic aerosols.

**Blue text below is our response to the referee's comments (reproduced in black). Corresponding new line numbers in revised manuscript are given in red.**

**Referee 1**

Zhao et al. apply idealised aerosol scenario experiments to examine the importance of aerosol forcing on mPWP tropical precipitation and the possibility of aerosol uncertainty explaining the mismatch between reconstructions and simulations. Based on these simulations, the authors suggested that i) removal of aerosol pollutants cause global warming and further warms the Northern Hemisphere due to asymmetrical emissions; ii) the Intertropical Convergence Zone (ITCZ) becomes narrower and stronger and shifts northward after removal of anthropogenic aerosols. The content of this manuscript is potentially interesting and there is certainly publishable material in it. However, the manuscript is far from being acceptable to an international journal in its current form, as it has many scientific uncertainties for the reader. Some interpretations are self-contradictory and needs to be untangled. My detailed comments are listed as below.

**We would like to thank the reviewer for the comments and are happy to make the revisions as suggested.**

Line 51, the authors stated that "mPWP simulations now use modern-day or pre-industrial aerosol concentration that may differ from the conditions during the mPWP. It implies that aerosol effects may be one of the possible explanations for the

mismatch between reconstructions and simulations". However, in the experimental design, the authors also used the modern-day and pre-industrial aerosol concentration, and explained the mismatch between reconstructions and simulations based on these simulations, sinks into logic antinomy.

**Thank you for raising this question. A brief explanation is that the highlighted sentences say that simulated mPWP climate change does not include aerosol forcing, while our analysis mainly focuses on aerosol effects in mPWP simulations. The first highlighted sentence is trying to say that traditional mPWP simulations usually set prescribed aerosol concentrations same as the control runs, which means that they do not consider aerosol forcings. Our work uses three simulations: a pre-industrial control simulation (PI), a simulation with mPWP boundary condition plus pre-industrial aerosol (Plio_Pristine) and one with mPWP boundary condition plus industrial pollutants (Plio_Polluted). The simulated mPWP change from the highlighted sentence is equivalent to "Plio_Pristine - PI" in our study (Sect. 3.1). We then compares the simulated mPWP change (Plio_Pristine - PI) with the reconstructions in this section, in purpose of firstly evaluating our simulated mPWP climate and secondly pointing out the existence of data-model mismatch. Plio_Pristine - Plio_Polluted considers effects of aerosol forcing in the mPWP simulations (Sect. 3.2 and 3.3). Idealised aerosol scenarios were chosen, because the realistic mPWP aerosol concentrations are not available. Plio_Polluted uses industrial pollutants (Lamarque et al., 2010), which is a real existed aerosol forcing though not happened during the mPWP. Analysis related to aerosol effects does not touch data-model comparison. We have rewritten it to "The discrepancy between mPWP simulations and reconstructions (Haywood et al., 2013, 2020) implies that models might miss some important mechanisms (Fedorov et al., 2013) or prescribed forcing could be a source of uncertainty (Feng et al., 2019). The mPWP simulations now use modern-day or pre-industrial aerosol concentration same as the control runs that may differ from the conditions during mPWP. Lack of including the effect of aerosol forcings and the usage of unrealistic prescribed aerosol concentration in mPWP simulations implies that aerosol effects may be one of the possible explanation for the mismatch between reconstructions and simulations."****(new line 53)**

The authors explained the spatial pattern of temperature change using asymmetrical emissions. However, the atmosphere is a dynamic collection of gases that constantly move and change, so the aerosols in the atmosphere will distribute globally. How can the aerosols keep asymmetrical distribution? The authors should give more discussion.

**Plio_Pristine and Plio_Polluted simulations in our study use prescribed aerosols, which give aerosol distribution instantaneously at the beginning and then left the model to respond to it. Aerosols do not distribute globally as the lifetime of aerosols is only a few weeks. The atmosphere is not well-mixed and the North-South difference emissions persists into concentrations, so the aerosols keep asymmetrical distribution.**

Lines 169-171, these two sentences are contradictory, please check if the description is incorrect. Based on fig. 1a and fig. 5a, I think the mPWP boundary conditions cause more warming than removing anthropogenic aerosols.

**The mPWP boundary conditions cause more warming than removing anthropogenic aerosols. That's what conclude in line 171 as "...This suggests that changes in boundary conditions are relative more important than aerosol effects**

[Figure]

**Figure 1. (Fig.7 in revised manuscript) Zonal mean changes in (a) vertical velocity ($10^{-3}$ Pa s$^{-1}$) and (b) meridional stream function ($10^9$ kg s$^{-1}$) by removing anthropogenic from the mPWP atmosphere (Plio_Pristine - Plio_Polluted).** Overlaid contours show the climate mean for the Plio_Pristine simulation (solid for positive values and dashed for negative values).

in mPWP warming." We are trying to say that removing aerosols only causes more over Northeastern Pacific and high-latitude North Atlantic Ocean than the mPWP boundaries. We have revised the sentence to "However, the mPWP boundary conditions causes more warming than removing anthropogenic aerosols nearly global except Northeastern Pacific and high-latitude North Atlantic Ocean (Fig. 5a,c,e). More warming lead by the mPWP boundaries than removal of aerosols suggests ..." (new line 181) to emphasise that the mPWP boundaries lead to more warming than removal of aerosols.

Line 176, the author stated that the ITCZ becomes narrower. However, I think it is really hard to get this conclusion only based on the precipitation changes. The authors should provide more convincing evidence.

We have added a figure (new line new Fig.7) and discussion (new line 198) about changes in the Hadley circulation.

Lines 180-183, the authors stated that "The pattern of the magnitudes of tropical precipitation anomaly matches the tropical temperature warming over oceans". I am unable to obtain this result from Fig 5. For example, the authors stated that tropical precipitation increases greater over the tropical Pacific and Indian Ocean, however, the temperature changes are similar over the whole tropical oceans.

The temperature along the Equator in the Pacific Ocean is greater than surroundings. The warming band is largely overlap with the precipitation increase belt.

Lines 193-197, the sentence "Land regional......cooling effect" is hard to be understood. The first half of the sentence expresses that the monsoon are enhanced, but the second half shows that monsoon are weakened. Please rewrite.

**The evidence (the citations in sentences) in the second half says that the cooling effect induced by increased aerosol during 1950s-1980s and over the 20th century lead to monsoon reductions. Logically, monsoon would be enhanced in the opposite situation (i.e. reducing aerosols), which matches the first half sentence saying that removing aerosols result in monsoon enhancement. We have added this logic in text by revising the sentence to "As anthropogenic emission partly caused the global land monsoon precipitation reduction during 1950s to 1980s and weakened western African and eastern and southern Asian monsoon over the 20th century by cooling effect (Douville et al., 2021), reducing aerosol-induced cooling would increase the precipitation over western Africa and eastern and southern Asia during local monsoon seasons. In our case, land regional monsoons are enhanced after removing anthropogenic aerosol, which is consistent with the expectation." (new line 208)**

Lines 218-220, the sentence is self-contradictory. The authors stated Plio_Pristine - Plio_Polluted have relatively small effects on surface wind in the first half of the sentence, then stated that removing anthropogenic aerosols have much stronger effects on surface wind.

**Sorry we made a mistake here. We have rewritten the sentence to "Over tropical and subtropical regions, the overall uniform warming induce by mPWP boundaries (Plio_Pristine - PI) result in general decrease in sea level pressure and have relatively small effects on surface wind (Fig. 10a,c,e). Though removal of anthropogenic aerosols also show warming in tropical and subtropical regions, it rises the sea level pressure and have much stronger effects on surface wind that shows seasonal variances (Fig. 10b,d,f)." (new line 233)**

Technical corrections

Please indent the first line of each paragraph, which makes easier for readers to read.

**We have added space in front of paragraphs.**

Replace "0 - 0.8 °C" with "at 0-8 °C", otherwise, it's hard to distinguish with minus sign. Please check the whole paper and correct it.

**Fixed. (new line 1, 25, 177, 185, 186)**

In the Model description, please add some sentences to introduce the relation between the CESM and CCSM4. Moreover, add the full name of CCSM.

**CESM1 is an updated version of CCSM4. CCSM4 (Gent et al., 2011) was developed by the US National Center for Atmospheric Research (NCAR) along with contributions from the academic community. CCSM4 were developed into an ESM, i.e. CESM1 (Hurrell et al., 2013), by adding capabilities related to BGC processes and aerosol effects. We have given the full name of CCSM4 and the relationship between the two models by "CESM1 was developed from the Community Climate System Model version 4 (CCSM4, Gent et al., 2011) by adding capabilities related to BGC**

115 **processes and aerosol effects (Meehl et al., 2013)." (new line 83)**

line 175, add reverse parentheses "A2)".

**Revised. (new line 187)**

120 Please improve the Figures. For example, i) delete the grid in the figures; ii) In figure 1, the "surface temperature change (°C)" coincided with "(b) PR Plio_Pristine – PI".

**Figures have been replotted to improve their quality.**

**Referee 2**

125 Comments on "Aerosol uncertainties in tropical precipitation changes for the mid-Pliocene Warm Period"

In this study the authors analysed the climate uncertainties resulted from aerosol scenarios in the simulations of mid-Pliocene Warm Period. Three simulation experiments PI, Plio_Pristine, and Plio_Polluted were analysed, which represent pre-industrial, mid-Pliocene with pre-industrial emissions, and mid-Pliocene with pre-industrial emissions plus industrial pollutants. They found that the change of precipitation between Plio_Pristine and Plio_Polluted is generally larger than that

130 between Plio_Pristine and PI. And they concluded that the aerosol forcing was the dominant driver in tropical precipitation change during mid-Pliocene.

This work seems to be a continuation of the work in Feng et al. (2019) with further analysis of the model results. In my opinion, there are some novel findings but not enough and explicit under current analysis. The details of some results and discussion are missing. And the conclusion is too strong. So I suggest the authors to make a major revision.

135 The general comments and specific comments are listed below:

**Thank you for commenting our manuscript. We are happy to adopt many of the comments and have revised the manuscript as requested.**

General comments

140 1. Please reduce the number of colors used in the colorbars in the figures, it is difficult to relate the colors with the value ranges.

**Figures have been replotted.**

2. With the model experiments designed or shown in this study, it is reasonable to study the aerosol uncertainties with two

145 idealised aerosol scenarios. However, the conclusion (aerosol forcing is the dominant driver in tropical precipitation change in mid-Pliocene) can not be derived following this work. First, in the PlioMIPs, the emissions of aerosols and their precursors were designed to be following the pre-industrial configuration (Haywood et al., 2011), so adding the industrial pollutants is

an idealised experiment and not based on proxy data or derived mid-Pliocene conditions. Therefore, you can not say aerosol forcing was the dominant driver in mid-Pliocene, because the forcing was not real. Secondly, you may need more experiments to compare individual components. For example, an experiment with only aerosol change but keeping CO2 in a pre-industrial level, an experiment without aerosol-cloud interaction, etc. You can refer to the experiments in Feng et al. (2019) and Sagoo and Storelvmo (2017).

We appreciate that the aerosol forcing that we investigated are idealised, and therefore cannot hope to quantify the amount of uncertainty in Pliocene aerosol. However, we have demonstrated that it is could as large that resulting from other boundary conditions, in certain variables such as tropical rainfall. We had aimed to convey aerosols can be the dominant source of uncertainty, rather than the dominant driver. It is clear from the reviewers' comments that we have not made that as clear as we could have. In a revised manuscript, we have rephrased the conclusions to draw out this distinction by revising the end of Conclusion to " Though we investigate idealised aerosol forcing on mPWP climate that cannot quantify the magnitude of uncertainty in Pliocene aerosol, we still argue that aerosols can be a dominant source of uncertainty as it could contribute to change in certain variables, such as tropical rainfall discussed in this study, as large as from other mPWP boundary conditions. It is important to consider the differences in aerosol conditions across the periods." (new line 289)

We are not aware of any research that aerosol forcing and the resulted climate responses are state-dependent to the first order. This means that comparing Plio_Pristine with Plio_Polluted should yield the same results to comparing PI_Pristine and PI_Polluted as the later is equivalent to our comparison but at a lower CO2. Comparison between our results and experiment with only aerosol changes but keeping CO2 in the preindustrial level would address the issue of whether the effect of aerosols is dependent on prescribed CO2. This is point is very interesting, but outside the scope of this study. An experiment without ACI would be CCSM4 experiments with Plio-Pristine and Plio-Polluted aerosol scenarios. However, we don't have the computing resource to run those simulations. We want to point out that the direct aerosol forcing without ACI is very small (Myhre et al., 2013). Having those simulations would not change our results and conclusion. Furthermore, this would be similar to the comparison between CCSM4-PlioMIP1 and Plio_Pristine with caveat that many model parameterizations have to be changed to simulate ACI, which inevitably changes model sensitivity and simulated moist state.

Specific comments

P1, L4: 'is' -> 'it'

Fixed.(new line 4)

P2, L51-53: "mPWP simulations now use modern-day or pre-industrial aerosol concentration that may differ from the conditions during the mPWP. It implies that aerosol effects may be one of the possible explanation for the mismatch between reconstructions and simulations." The logic is not obvious here between two sentences. Please clarify it or add references.

The highlight sentences were trying to point out that the mPWP simulations might be improved if use more realistic aerosol, as the simulations usually use prescribed aerosol concentration same as the pre-industrial control run. We have rewritten it to "The discrepancy between mPWP simulations and reconstructions (Haywood et al., 2013, 2020) implies that models might miss some important mechanisms (Fedorov et al., 2013) or prescribed forcing could be a source of uncertainty (Feng et al., 2019). The mPWP simulations now use modern-day or pre-industrial aerosol concentration same as the control runs that may differ from the conditions during mPWP. Lack of including the effect of aerosol forcings and the usage of unrealistic prescribed aerosol concentration in mPWP simulations implies that aerosol effects may be one of the possible explanation for the mismatch between reconstructions and simulations." (new line 53)

P3, L62-63: "This increases the shortwave cloud radiative forcing, thus, cools surface temperature and amplifies polar warmth (Sagoo and Storelvmo, 2017)." Please check the reference, I think if you have cooling effect on the surface, the polar amplification means the amplified polar cooling.

Yes, you are right. Simulations of Sagoo and Storelvmo (2017) suggest high dust result in surface temperature cooling and polar amplification (from their high dust and LGM simulations) and low dust result in surface temperature warming and polar amplification (from their low dust and mPWP simulations). "Results showed that increased dust reduces the size of ice crystals in clouds while increasing their amount. This increases the shortwave cloud radiative forcing, thus, cools surface temperature and amplifies polar warmth (Sagoo and Storelvmo, 2017)." has been revised to "Their mPWP simulation modified by extreme low dust suggests surface temperature warming and polar amplification due to reduced radiative forcing by increased size while decreased amount of ice crystals in clouds (Sagoo and Storelvmo, 2017)." (new line 65)

P3, L69-70: "potential analogue nature of Pliocene climate, not to the present-day polluted climate, but to future climate scenarios which feature removal of anthropogenic pollutants" Which future scenario do you refer to here? Please specify it.

Most scenarios for future projections in the IPCC AR6 have a decrease in the anthropogenic aerosol forcing, which will contribute to a increase in global mean annual surface temperature and precipitation (Lee et al., 2021). We have added the citation of Lee et al. (2021) in text.

P3, L77: Write the full name of CCSM4.

We have given it's full name the Community Climate System Model version 4 (CCSM4, Gent et al., 2011) in text. (new line 83)

P3, L79-82: "which uses Aiken, accumulation and coarse modes to solve number and size concentration of internal condensation and coagulation of different species among modes." What are "size concentration", "internal condensation"? This sentence is not clear and needs to be rephrased.

**That should be size and number concentration instead. MAM3 solves for sizes and number concentrations of internal mixtures of dust, sea-salt, sulfate, organics and black carbon using three categorical size bins of 0.02–0.08 $\mu$m (Aiken mode), 0.08–1 $\mu$m (Accumulation mode) and 1.0 – 10 $\mu$m (Coarse mode). We have rewritten the sentence to "... which uses Aiken (0.02–0.08 $\mu$m), accumulation (0.08–1 $\mu$m) and coarse (1.0–10 $\mu$m) modes to solve size and number concentration of internal condensation and coagulation of different species among modes." (new line 88)**

P3, L87: Please list key configurations or parameters in CCSM4-PlioMIP1 simulation related to this work.

**The CCSM4-PlioMIP1 simulation was generated by CCSM4, an earlier version of CESM. Key improvements between the two generations have been described Meehl et al. (2013) in as well as in Sect. 2.1. Prescribed mPWP boundary condition should have no effect on simulations, as the CCSM4-PlioMIP1, Plio_Pristine and Plio_Polluted all used PlioMIP1 boundary condition (Haywood et al., 2010; Dowsett, 2007). Plio_Pristine and Plio_Polluted were branched from the CCSM4-PlioMIP1 simulation, because it would save time and computing cost of letting model reach its equilibrium state. CESM would only need to response to change in aerosol forcing. Therefore, we feel there is no need to list key configurations or parameters of CCSM4-PlioMIP1 simulation.**

P3, L87: Does 'direct effect' mean 'aerosol direct effect'? Please clarify it.

**Yes. "aerosol" has been added.(new line 95)**

P3, Sec. 2.2: Are the experiments the same as the ones in Feng et al. (2019)? From the text in P3, L66-67, it seems that the simulations are the same. If they are the same, did you directly use the output data from their experiments or you rerun the experiments? Please specify it clearly in the text.

**Yes, simulations are the same, which is mentioned in Line 66-67 "The simulations were performed initially to analyse ..." We have revised the sentences to "In this study, we analyse two Pliocene simulations that were performed initially to analyse the effects of aerosol-cloud interactions on mPWP seasonally sea ice-free Arctic (Feng et al., 2019) to further investigate the potential effect of aerosol on mPWP climate. Different aerosol scenarios were applied, one with pre-industrial aerosol concentrations and one with present-day aerosol concentrations published in Lamarque et al. (2010)." We also emphasise it again in Sect. 2.2 by adding "The existed simulations (Feng et al., 2019)..." at the beginning.(new line 94)**

P4, L90-92: "The other prescribed pre-industrial emissions plus industrial pollutants of anthropogenic SO2 , sulfate and organic compounds estimated for the 2000s from an gridded (0.5° x 0.5°) emission dataset (hereafter referred as to Plio_Polluted) published in Lamarque et al. (2010)." The description of the configurations of emissions are not clear. For example: (1) Does "industrial pollutants of anthropogenic SO2" mean the SO2 emissions from industry sector, excluding all the emissions from other sources like vehicle emissions? (2) Are other anthropogenic emissions included here, like NOx, black carbon? (3) Here

250      SO2 could be the precursor of aerosols, and sulfate and organic compounds can be directly emitted or formed from precursors, please provide more details of your emission configurations.

    **Lamarque et al. (2010) published a dataset of historical (1850-2000) anthropogenic (defined as originating from industrial, domestic and agriculture activity sectors) and biomass burning emissions of reactive gases and aerosols. The dataset provided consistent gridded emissions for CMIP5 models to run chemistry model simulations that contributed**

255     **to the assessment in IPCC AR5. The emissions for year 2000 of the dataset, which prescribed in Plio_Polluted, serve as an anchor point for historical emissions and future emissions Lamarque et al. (2010). (1) "industrial" is relative to "pre-industrial" ahead, which refers to time instead of a source. (2) No, NOx and black carbon were excluded. Emissions were constant in the Plio_Pristine and Plio_Polluted. (3) Lamarque et al. (2010) is an emission dataset, i.e. from primary source. Secondary sources are not included. Emitting configuration is well documented in Lamarque et al.**

260     **(2010) as cited in text. We have added a bit more to introduce the dataset by "... published in Lamarque et al. (2010). The emission dataset provided consistent gridded anthropogenic (defined as originating from industrial, domestic and agriculture activity sectors) and biomass burning of reactive gases and aerosols covering the historical period from 1850 to 2000 CMIP5 models to use in running chemistry model simulations that would contributed to the assessment in IPCC AR5 (Lamarque et al., 2010)." (new line 100)**

265

     P4, L110: "PI" is used before defined. And please also specify what PI simulation represents, e.g., an average over a period from a specific CMIP5/6 simulation experiment?

    **PI has been defined here. (new line 121) In modelling, PI usually refers to the pre-industrial control run, which applies pre-industrial (1850 CE) boundary condition and performs as a baseline for computing anomaly. In this study,**

270     **PI refers to a simulation run by CESM model with pre-industrial boundary condition prescribed. We think there is no need to specify it in text.**

     P5, L122-123: "The combination regenerates the site sets 35, 72; 22, 77; 79,61; 58,61,79,101 and 65, 66 into new sites located at the center of sets." Where do the numbers come from? Please clarify it.

275     **Site numbers come from the the compilation of mPWP proxy records of Feng et al. (2022). We have added its source in text, as "The combination regenerates the site sets 35, 72; 22, 77; 79,61; 58,61,79,101 and 65, 66 of Feng et al. (2022) into new sites located at the center of sets." (new line 134)**

     P5, L131-133: "The underestimation in Northern Hemisphere warming may be explained by that Plio_Pristine was branched

280 from an earlier CCSM4-PlioMIP1 simulation that had underestimated the warming in the Northern Hemisphere (Rosenbloom et al., 2013)." What about the results from the latest simulations in PlioMIP2? It would be better to also compare your results with the newer experiment results.

    **First of all, the simulations used PlioMIP1 boundary conditions. Secondly, though PlioMIP2 simulations produce warmer and wetter mPWP than PlioMIP1, the difference is caused by the addition of new and more sensitive models**

285 **instead of the modifications in boundary conditions between the two PlioMIPs (Haywood et al., 2020). CESM's earlier version CCSM4 participated in both PlioMIPs and show that PlioMIP2 anomalies are similar to PlioMIP1's (Haywood et al., 2020). Therefore, it would be appropriate to compare with the PlioMIP1 outputs here. Comparison with the PlioMIP2 results would not be necessary.**

290 P5, L134-135: "13 out of the 37 sites show a mismatch between simulated surface temperature anomaly and reconstructed SST anomaly smaller than 1.0°C." Please also add the reference here besides in the figure caption.

**Foley and Dowsett (2019) has been added. (new line 146)**

P5, L138-139: "The reconstructed zonal temperature gradient along the tropical Pacific Ocean is around 0.5°C greater than
295 simulations (Plio_Pristine - PI)" How do you define the zonal gradient here, is it the temperature difference between two regions? How the regions are defined?

**The reconstructed temperature gradient is calculated as the temperature difference between sites near Colombia and sites near SE Asia. We give the regions and change "zonal" to "meridional" for being easier to understand. The sentence has been revised to "The reconstructed meridional temperature gradient along the tropical Pacific Ocean**
300 **(between sites near Colombia and sites near Southeast Asia) is around 0.5°C greater than simulations (Plio_Pristine - PI)." (new line 149)**

P5, L147-148: "Precipitation decreases over subtropical oceans, eastern parts of South America and monsoon areas over North America." The proxy data in Fig. 1b show enhanced precipitation almost the whole globe, with only few sites showing reduced precipitation, which indicates a global scale of the precipitation increasing and a regional/local scale of the decreasing.
305 However, the simulation results show reduced precipitation over a large area, especially the Atlantic Ocean. Could you explain the reason, or could you find any references which mentioned it or tried to explain it?

**The spatial pattern of the increased precipitation over low and high latitudes and the reduction over part of the parts of the subtropics in Fig.1b is consistent with the PlioMIPs results (Haywood et al., 2013, 2020). In the mPWP simulations, precipitation occurs more in the tropics than the subtropics by having a weakened Hadley circulation that is**
310 **related to decreased zonal temperature gradient between the tropics and the subtropics (Corvec and Fletcher, 2017). Models also produce a poleward shift in mid-latitude westerlies during the mPWP, which is linked to reduced meridional temperature gradient (Li et al., 2015). The poleward shift in mid-latitude westerlies results in a poleward shift in higher-latitude precipitation. The precipitation over subtropics, thus, shows a reduction in the mPWP simulations.**

P6, L155: "Reducing atmospheric aerosols is expected to warm the climate." It needs more details and references.
315 **Aerosols that mainly scatter solar radiation have a cooling effect, and those absorb solar radiation have a cooling effect. Scattering and absorbing aerosols mix in the atmosphere. Net effect of aerosols on energy budget is dependent on the characteristics of surface and cloud. According to IPCC AR6, human-induced aerosols contributed to a cooling of 0.0°C to 0.8°C between 2010-2019 relative to 1850-1900 (IPCC, 2021b). Therefore, removing anthropogenic aerosols is expected to warm the climate. We have added "According to IPCC AR6, human-induced aerosols contributed to a**

320 **cooling of 0.0°C to 0.8°C between 2010-2019 relative to 1850-1900 (IPCC, 2021b). Therefore, removing anthropogenic aerosols is expected to warm the climate." in text.** (new line 165)

P6, L169-171: I do not understand the logic of these two sentences, please clarify it.

**We are trying to say that removing aerosols only causes more over Northeastern Pacific and high-latitude North**
325 **Atlantic Ocean than the mPWP boundaries. We have revised the sentence to "However, the mPWP boundary conditions causes more warming than removing anthropogenic aerosols nearly global except Northeastern Pacific and high-latitude North Atlantic Ocean (Fig. 5a,c,e). More warming lead by the mPWP boundaries than removal of aerosols suggests ..."** (new line 181) **to emphasise that the mPWP boundaries lead to more warming than removal of aerosols.**

330 P6, L176: Why do you think ITCZ is narrower in your simulation Plio_Pristine compared to Plio_Polluted? Please clarify it.

**We have added analysis on change in Hadley Circulation (Fig.7 in revised manuscript). Sentences have been revised to "Removing anthropogenic aerosols leads to stronger vertical velocity at the Equator (Fig. 1a). The meridional circulation in the tropics is strongly enhanced in the Southern Hemisphere and slightly weakened in the Northern Hemisphere (Fig. 1b) after removing anthropogenic aerosols from the atmosphere, which is related to a northward shift of the ITCZ.**
335 **The narrower ITCZ is consistent with a warmer climate simulated by Plio_Pristine, and agrees with an earlier study which suggests that future ITCZ would become narrower and weaker under global warming in the CMIP5 simulations (Byrne and Schneider, 2016). Ridley et al. (2015) and Voigt et al. (2017) found that introducing anthropogenic aerosols causes southward shift of the ITCZ. In opposite, removing anthropogenic emission would expect a northward shift of the ITCZ. Our results are consistent with this finding."** (new line 188)

340

P6, L183-184: "consistent with the positive correlation between tropical precipitation change and spatial deviations of SST warming from the tropical mean (Xie et al., 2010)" Then what is the reason of decreasing precipitation over the warming SST areas in Fig. 5?

**We focus on annual change here, which shows an overall enhancement along the tropics that is largely affected by the**
345 **strong increase during JJA. If you mean the decreasing precipitation over the tropics during DJF, a possible explanation is due to the change in low-latitude easterlies (Fig.9d), as the low-latitude easterlies is strengthened between 30°N(S) and roughly 10°N(S) while is weakened between 10°N(S) over the tropical Pacific.**

P7, Sec. 4.1: The discussion in this section about interaction of aerosol and cloud is not clear. (1) It is better to first give the
350 definitions of net shortwave/longwave flux, clear-sky/full-sky net shortwave/longwave flux, shortwave/longwave cloud forcing, and also how you estimate the aerosol direct effects from these variables. In Fig. 3, the aerosol direct effect is not shown, so you need to clarify it.

**The original Fig.3a plots change in net shortwave and longwave flux at top of model under full-sky and clear-sky condition, respectively. At top of model can be viewed as at top of the atmosphere (TOA). Net shortwave/longwave flux are**

355 **TOA shortwave absorption and longwave emission. clear-sky/full-sky defines the cloud status, i.e. full-sky means cloudy sky and clear-sky refers to no clouds. For example, clear-sky net shortwave flux means TOA shortwave absorption with no clouds in the atmosphere. Clear-sky variables show aerosol direct effect. Difference between net full-sky (cloudy sky) and clear-sky (no clouds) flux would show aerosol indirect effect, which can be understood as cloud forcing. For a better understanding, we have replaced "cloud forcing" by "cloud radiative effect" defined as "The radiative effect of**
360 **clouds relative to the identical situation without clouds" by IPCC (IPCC, 2021a). Figure has been regenerated, which now plots aerosol direct and indirect effects. We have given the definition of these fluxes in the revised figure caption.**

(2) "while clouds contribute to a net forcing greater than 2 W m-2 with more shortwave absorption and less longwave emission" Which latitudes do you mean here? In this phrase "more shortwave absorption and less longwave emission", do you
365 mean on the surface or in the cloud level?

**The "2 W m-2" comes from tropical average (23.5°N to 23.5°S), which can be seen in Fig.3b." more shortwave absorption and less longwave emission" here refers to change in cloud radiative effect at top of model. According to the definition of cloud radiative effect and the properties of clouds, we have revised "more shortwave absorption" to "less shortwave reflection". This section has been rewritten.**

370

(3) "is still less important than the change in shortwave cloud forcing." How can you find this from Figs. 3 and 4?

**Aerosol affect energy flux through direct effect by reflection or indirect effect through interaction with clouds. Difference between net full-sky (cloudy sky) and clear-sky (no clouds) flux would show aerosol indirect effect, which is equivalent to cloud foricng. Shortwave cloud forcing is measured by subtracting TOM downwelling shortwave radia-**
375 **tion of clear-sky condition from the full condition. Fig. 3 has been replotted to clarify the change, which now shows aerosol indirect effect is more important here.**

(4) "Figure 6 shows decrease in droplet concentration (especially over lands and high latitudes, Fig. A3a) and decrease in cloud liquid path (except tropical Pacific between 5°S and 5°N and western Africa, Fig. A3b) after the removal of pollutants."
380 How could you conclude this from Fig. 6?

**We do apologise. There was extra panels about annual averaged change in droplet concentration and cloud liquid path in our old Fig.6. We feel Fig A3 would be sufficient, so we removed these panels before submission. We have revised this sentence to "Removal of pollutants decreases droplet concentration (especially over lands and high latitudes, Fig. A3a) and decrease cloud liquid path (except tropical Pacific between 5°S and 5°N and western Africa, Fig. A3b)." (new**
385 **line 228)**

(5) The title of this section is "Key forcing driving tropical precipitation change", but there is no discussion about how the forcings are driving tropical precipitation change. Please add more discussion details.

[Figure]

**Figure 2. (Fig.3 in revised manuscript) Zonal averaged change in (a) shortwave radiation flux and (b) cloud radiative effect in W m$^{-2}$ at top of model (TOM) after removal of anthropogenic emissions (Plio_Pristine - Plio_Polluted)**. TOM radiation fluxes are used to show change at top pf atmosphere (TOA), because the results are not sensitive to TOM or TOA. (a) Clear-sky shortwave radiation (orange) refers to TOM radiation flux in the absence of clouds, and full-sky shortwave radiation flux (black) is with clouds. Shortwave cloud radiative effect (cyan) is the difference in shortwave radiation flux between full-sky and clear-sky conditions. (b) Could radiative effect (black) is the sum of downwelling shortwave (green) and longwave (purple) could radiative effect.

**Analysis in Sect. 3 shows that tropical precipitation enhancement after the removal of anthropogenic emissions is due to the further warming by removing aerosol-induced cooling from the atmosphere. The aim of this section is finding out which aerosol effect (direct/indirect) is more important. The first half discussion shows that aerosol radiation forcing is less important than cloud forcing, i.e. aerosol indirect effect is more important. The second half shows that removing anthropogenic emissions affects cloud forcing by reducing cloud albedo and decreasing cloud lifetime. We have renamed this section as "Key forcing contributing to aerosol effect".**

P8, L225-228: "Figure 10 shows the relative importance of removing anthropogenic aerosols and mPWP boundary conditions on mPWP zonal mean precipitation change. The ratio (panel a) indicates that the relative importance of effects of aerosol forcing and mPWP boundary conditions (including high CO2) on precipitation is complicated, but overall aerosol effect is more important over the tropics (panel b), which could imply the importance of aerosol scenario in simulating Pliocene climate." (1) Fig. 10a is difficult to see the features, please find a better way to plot the data, maybe increase the width-height ratio?

**We have replotted the figure, which now is Fig.11 in the revised manuscript.**

(2) Please add more detailed discussion about the relative importance instead of just one word "complicated".

**"complicated" comes form the the ratio of change in latitudial averaged annual, JJA and DJF precipitation change shown in Fig.10a whose patterns are complex. We are trying to detect which (aerosol forcing or mPWP boundary condition) is more important but not why. Overall, Plio_Pristine - Plio_Polluted shows greater change in tropical zonal averaged annual mean precipitation than Plio_Pristine - PI, as shown in new Fig.11. We have revised the sentences to "Figure 11 shows the relative importance of removing anthropogenic aerosols and mPWP boundary conditions on mPWP zonal mean precipitation change. Precipitaion change in reponse to aerosol forcing and mPWP boundary conditions (including high $CO_2$) on precipitation shows complicated pattern as there is no uniform changing trend."** **(new line 240)**

(3) It is difficult to see "aerosol effect is more important over the tropics" from Fig. 10b, maybe a global map of the ratio is better to represent it?

**Before hand, we apologise that we made a mistake in caption, as dark markers actually show latitudal-averaged change within the tropics and the light colors for those outside of tropics. We have replotted Fig. 10 as the new Fig.11 in revised manuscript.**

**Our analysis is focusing on annual change. The reason of giving seasonal change is to show if the annual change is largely affected by seasonal variance.**

[Figure]

**Figure 3. (Fig.11 in revised manuscript) Effect of removing anthropogenic aerosols and mPWP boundary conditions on mPWP zonal mean precipitation change and their relative importance.** Left column shows zonal averaged (a) annual, (c) DJF and (e) JJA mean precipitation change induced by removing anthropogenic aerosols (dashed) and mPWP boundary conditions (solid). Left column shows ratio of change in (b) annual, (d) DJF and (f) JJA precipitation change computed as ((Plio_Pristine - Plio_Polluted) / (Plio_Pristine - PI)). Yellow shading highlights where -1< ratio < 1.

P8, L241-242: "SO2 increased by 11 Tg in Unger and Yue (2014) mPWP simulations, yet SO2 from anthropogic sources is estimated to be 120 Tg from 1980 to 2000s (Smith et al., 2011)." Here 11 Tg should be 11 Tg/yr, which is the emission rate, the same for 120 Tg/yr.

**We have corrected the unit. (new line 257)**

P8, L42-43: "As such, despite that enhanced biogenic emissions may compensate some of the responses seen in the Plio_Pristine" For "compensate", do you mean: "If the enhanced biogenic emissions, which will increase the secondary organic aerosol formation and growth, is included in the Plio_Pristine simulation, the difference between Plio_Pristine and PI will decrease."? The statement here is not clear, please clarify it.

**Sect. 4.3 is for implications for simulating Pliocene climate, not only focusing on the results of this study. According to sentences before Line242-243, though mPWP was expected to have stronger biogenic emissions from more active wildfire and biome productions, the mPWP emission is likely much smaller than the anthropogenic emission during the historical period (1850CE onwards till now) and near future. "compensate" here means the reduced difference between mPWP and near future climate. Because the compensation is small, we make an argument in the following half sentence as "we argue that the absence of anthropogenic pollutant from the mPWP troposphere remains one of the important factors driving the differences between the mPWP and near future climate, especially the Arctic climate (Feng et al., 2019) and hydroclimate."**

P8, L246-247: "Sagoo and Storelvmo (2017) looked in to the effect of dust and demonstrated that increased dust emissions resulted in an enhanced dynamical response that lead to great changes in hydrological circulations." The dust emission in mPWP is lower compared to pre-industrial, why you are talking about "increased dust emissions"?

**This sentence is about Sagoo and Storelvmo (2017)'s major finding based on their high-dust simulations, rather than their mPWP simulation. Their mPWP simulation used low-dust scanario. The sentence has been removed.**

P8, L247-249: "Their Pliocene simulation with idealised dust scenario shows greater change in precipitation than only consider CO2 forcing, yet only used idealized scenario (Sagoo and Storelvmo, 2017)." Please add more details, e.g., which scenario, how large is the "greater change in precipitation", global or tropical precipitation.

**We have rewritten the sentence to "Sagoo and Storelvmo (2017)'s Pliocene simulation with with idealised extreme low dust scenario shows greater change in precipitation than only consider $CO_2$ forcing, yet only used idealised scenario". (new line 262)**

P9, L251-252: Please add at least some simplified definition of the moisture budget component terms.

**It refers to a commonly used moisture budget analysis (Trenberth and Guillemot, 1995) which decomposes the anomalous moisture budget into thermodynamic (change in moisture flux convergence from moisture change, i.e. specific humidity), dynamic (relates to change in wind with unchanged moisture) and residual (contributed by transient**

**eddy), and change in dynamic component is related to the pattern of anomalous net energy input that is affect by verti-cal integrated moist static energy. We have revised the sentence to "The anomalous moisture budget can be decomposed** in to thermodynamic (due to change in specific humidity), dynamic (due to change in mean circulation) and residual (due to transient eddy) components etc (Trenberth and Guillemot, 1995; Held and Soden, 2006)." (new line 266)

P9, L254-255: The statement from "high CO2 concentration" to "thermodynamic change plays an important role" is not explicitly clear here, please add more detailed discussion if you want to keep this statement.

**RCP8.5 scenaio is forced by high $CO_2$ concentration. L251-258 discussed which decomposed component of the anomalous moisture budget majorly drives the monsoon change in RCP8.5, i.e. forced by high-CO2 concentration. The decomposed components include thermodynamic component, dynamic component, residual component and net energy input component. We have revised the sentence to "D'Agostino et al. (2019, 2020) stated that monsoon changes under RCP8.5 are likely driven by thermodynamics and net energy input in response to global warming, which implies that high $CO_2$ concentration would affect precipitation by increasing the thermodynamic component of the moisture budget.. As the mPWP simulation also prescribed high atmospheric $CO_2$ concentration,...". (new line 268)**

P9, L255-257: Removal of anthropogenic emissions will also affect the thermodynamics itself in the atmosphere, and the anthropogenic emissions include both scattering (e.g., sulfate) and absorbing (e.g., black carbon) compositions, which are also related to shortwave and longwave radiation, so this process is complicated. Therefore, I do not think it "would logically increase the effects of thermodynamic and net energy input". Please add more discussion and references here.

**We agree that the process is complicated. However, black carbon is not included in our simulations. And as mentioned above, "thermodynamic and net energy input" here are the decomposed component of the anomalous moisture budget driving monsoon change.**

Figs. 6 and 8: Change between which cases? Please specify it in the figure caption.

**We have added "Plio_Pristine - Plio_Polluted" in captions to clarify the difference plotted in figures. The original Fig.8 is Fig.9 in revised manuscript.**

**Referee 3**

The climate in the mid-Pliocene Warm Period (mPWP) is usually viewed as an analog for future climate, but simulated mPWP climate change is largely different from the climate change in the 20th century. Also, there are large discrepancies between the simulation and proxies in the mPWP climate. To address the discrepancies, Zhao et al. suggested a different aerosol scenario from the pre-industrial aerosol setting should be used for simulating the mPWP climate. By removing pollutants of

490 the industrial period in the mPWP climate simulation, the authors get more warming in the Northern Hemisphere and stronger ITCZ. The manuscript is well-written and I suggest a minor revision. My comments are below.

**We would like to thank you for reviewing our manuscript and providing comments about our work. We are happy to adopt many of the comments and have revised the manuscript as requested.**

495 1. How the results of this study can help resolve the underestimation of temperature increase in the climate models should be addressed. In the abstract and Introduction, the authors mentioned discrepancies between models and proxies in the mPWP. One important difference is that the models underestimate the high temperature in the mPWP, especially in the high latitudes. When simulating the mPWP climate, people often use pre-industrial aerosol forcing, and the results of this study tell us that using pre-industrial aerosol forcing leads to increased warming. Thus, if we use more aerosol forcing, as the authors suggest

500 the mPWP may have more aerosols, the simulated mPWP climate should be even cooler, and the discrepancies between the simulation and proxies would be larger. This confuses me and I think it is important to let readers know how considering aerosols forcing scenario can help improve climate simulations in the mPWP.

**Thanks for raising the question, but there might be a misunderstanding here. Our results is opposite to your "Thus, if we use more aerosol forcing, as the authors suggest the mPWP may have more aerosols, the simulated mPWP climate**
505 **should be even cooler, and the discrepancies between the simulation and proxies would be larger" actually. Results of this study do not mention that using pre-industrial aerosol forcing leads to increased warming. Sect. 3.1 describes the warming driven by prescribed mPWP boundary conditions (Plio_Pristine - PI) and compares the simulated mPWP change with the reconstructions, which points out the existence of data-model mismatch. The mPWP climate is expected to have less aerosols than pre-industrial. Sect. 3.2 and 3.3 analyse the effect of removing anthropogenic aerosol emissions**
510 **(Plio_Pristine - Plio_Polluted). Results of these sections show that less aerosols lead to further warming. It means that if we use less aerosols, the simulated mPWP climate should be furhter warm, and the discrepancies between the simulation and proxies would be smaller.**

2. The title "Aerosol uncertainties in tropical precipitation changes for the mid-Pliocene Warm Period" confuses me. What
515 does "aerosol uncertainties in tropical precipitation" mean? Does it mean the uncertainties of aerosol bring uncertainties in tropical precipitation in climate models?

**"aerosol uncertainties in tropical precipitation" means the uncertainties induced by aerosol forcing in simulating (mPWP) climate.**

520 L28: Repeated "e.g."s
**Done.(new line 28)**

L39: "emission induced": What emission? GHGs or aerosols?

**It means aerosol emission. We have added "aerosol" in text. (new line 40)**

525

L118 and L121: (Feng et al., 2022) -> Feng et al. (2022)

**Fixed. (new line 130, 132)**

Figure 1: What are the circles in Fig. 1b? Do the proxies in Fig. 1b represent precipitation minus evaporation? If so, why not
530  compare them to the simulated P-E? Also, please increase the white space between Figure 1a and 1b.

**Circles in Fig.1b are reconstructed annual mean precipitation change published in Salzmann et al. (2008). Triangles are signals of hydroclimate change (precipitation minus evaporation) published in Feng et al. (2022). The aim of panel b is to show precipitation change rather than hydroclimate change. It is difficult to have abundant quantitative evidence of mPWP precipitation anomaly. Therefore, we choose to plot this new qualitative wet-dry dataset to give an idea of the**
535  **sign of precipitation change, though the compilation does not show precipitation anomaly directly. We have enlarged space between panels.**

L169-171: If "removing aerosols causing more warming than the mPWP boundary conditions", then why do you say that "the changes in boundary conditions are more important than aerosols?" This seems to contradict each other.

540  **We are trying to say that removing aerosols only causes more over Northeastern Pacific and high-latitude North Atlantic Ocean than the mPWP boundaries. We have revised the sentence to "However, the mPWP boundary conditions causes more warming than removing anthropogenic aerosols nearly global except Northeastern Pacific and high-latitude North Atlantic Ocean (Fig. 5a,c,e). More warming lead by the mPWP boundaries than removal of aerosols suggests ..." (new line 181) to emphasise that the mPWP boundaries lead to more warming than removal of aerosols.**

545

L212: Figure 6 is about droplet concentration. This should be Figure 7.

**We do apologise. There was extra panels about annual averaged change in droplet concentration and cloud liquid path in our old Fig.6. We feel Fig A3 would be sufficient, so we removed these panels before submission. We have rewritten this sentence to "Removal of pollutants decreases droplet concentration (especially over lands and high latitudes,**
550  **Fig. A3a) and decrease cloud liquid path (except tropical Pacific between 5°S and 5°N and western Africa, Fig. A3b)." (new line 228)**

L219: "general increase in sea level pressure": But the left panel of Figure 9 shows an increase in sea level pressure.

**Sorry we made a mistake here. We have rewritten the sentence to "Over tropical and subtropical regions, the overall**
555  **uniform warming induce by mPWP boundaries (Plio_Pristine - PI) result in general decrease in sea level pressure and have relatively small effects on surface wind (Fig. 10a,c,e). Though removal of anthropogenic aerosols also show warming in tropical and subtropical regions, it rises the sea level pressure and have much stronger effects on surface**

**wind that shows seasonal variances (Fig. 10b,d,f)." (new line 233)**

560     L227-228: I cannot see the "overall aerosol effect is more important" from Figure 10b. The numbers of dots above and below the 1:1 line are nearly equal.

    **Figure has been replotted.**

L270: due to mPWP had warmer and wetter climate -> due to warmer and wetter climate in the mPWP

565    **We have revised it. (new line 288)**